**communications** engineering

# A concentric tube catheter for endoluminal interventions, steered and imaged via magnetic resonance imaging
Peter Lloyd [1,3], Nikita Murasovs[1,3], Yael L. May [1], Alistair Bacchetti[1], Benjamin Calmé [1], Joshua Davy[1], Vittorio Francescon[1], James H. Chandler [1], Erica Dall'Armellina [2], Jurgen E. Schneider[2] & Pietro Valdastri [1] ✉

Two major challenges associated with robotic catheterization are, firstly, the provision of controllable degrees of freedom (DoFs) and, secondly, accessing feedback on the shape and pose of the catheter. Miniaturizable active steering can be achieved through magnetic actuation, and Magnetic Resonance Imaging (MRI) provides high definition, radiation-free 3D imaging that can be utilized for shape-sensing. Here, we propose a structurally adaptable Coaxial Sleeve Magnetic Actuator (CoSMA), with deformation energy provided by the background field of the MRI scanner. Our approach combines the magnetic actuation principle of the easy axis of alignment with the mechanical principles of concentric tube designs. This concept allows for a materially flexible ($E = \mathcal{O}(1\text{ MPa})$), and therefore risk reduced, multi-DoF catheter. We demonstrate the CoSMA, constructed of three coaxial components with respective outer diameters of 4 mm, 1.5 mm and 0.4 mm, in an aortic arch phantom navigation within the bore of a pre-clinical MRI scanner.

Soft active catheters have demonstrated their efficacy in accessing sensitive, restricted, and unstructured regions of the human anatomy[1,2]. Magnetically actuated devices have proven of particular interest due to a combination of material softness and extensive miniaturization potential[3]. Flexible Magnetic Catheters (FMCs) can provide a safer and more controllable pathway to otherwise inaccessible regions of the body offering clinical solutions not otherwise available[4]. In parallel to this, concentric tube robots (also known as active cannulas), offer high Degree of Freedom (DoF) deformation along an elastic continuum structure which can be fabricated at the millimeter scale[5] and have been shown to be MR compatible[6]. Concentric tube robots are made from several pre-curved tubes nested within one another and rely on material elasticity to transfer deformation energy from the operator to the robot. As such, they are typically made from Nitinol (Elastic modulus, E ≈ 50 GPa) which is stiff in comparison to living tissue. Furthermore, due to this inherent stiffness, these robots also suffer from the snap-through instability, where the robot can physically snap from one low energy configuration to another[7,8]. Concentric tube robots leveraging remnant magnetic actuation to enhance stability[9,10] or reduce stiffness[11,12] have been demonstrated but never in an MR compatible format. Here, we offer a fully hybridized approach combining the mechanical principles of concentric tube designs with the off-board energy provision of magnetic actuation to create a softer, and therefore safer, design, MR compatible by design, and with less susceptibility to the snap-through instability.

One of the major outstanding issues associated with the navigation of FMCs relates to device tracking, typically performed via fluoroscopy, which exposes both patient and operators to ionizing radiation whilst providing a low contrast 2D projection. Alternative sensing solutions to fluoroscopy do exist and have demonstrated encouraging results (at significantly lower cost than MRI) but are variously limited in terms of fidelity (e.g., ultrasound[13]), material stiffness (e.g., Fiber Bragg Gratings[14]) or, in the case of electromagnetic tracking (e.g., ref. [15]), provide singular pose feedback per sensor, imposing limits on both information availability and minimum physical size. MRI offers the current gold standard in non-ionizing high-resolution 3D imaging and thus, MR-compatible robotics (primarily, robots made of MRI-safe, non-ferrous materials) represents an active area of research with high potential[16].

MR actuated robotics represents an alternative paradigm to the aforementioned approaches where a robot is actuated and imaged by the MRI system[17]. This methodology has the potential to enable MR-guided procedures where diagnosis, surgery, and post-treatment assessment can all be performed in a single, integrated event. Research has been conducted in this field utilizing the imaging gradients (e.g., refs. [18–22]), the fringe field[23]

[1]STORM Lab, Department of Electrical and Electronic Engineering, University of Leeds, Leeds, UK. [2]ePIC Lab, Leeds Institute of Cardiovascular and Metabolic Medicine, University of Leeds, Leeds, UK. [3]These authors contributed equally: Peter Lloyd, Nikita Murasovs. ✉e-mail: p.lloyd@leeds.ac.uk

and the interaction between a controllable current through embedded micro-solenoids and the background field[24,25]. The challenges associated with the approaches used to date are the low magnitude and limited DoFs of force available via gradient coil actuation, the difficulties of miniaturization of embedded solenoids and the lack of imaging when utilizing the fringe field for actuation. Here we introduce our Coaxial Sleeve Magnetic Actuator (CoSMA) which leverages the phenomenon of the easy axis of magnetization coupled with mechanical control of position and orientation of coaxial sleeves (Fig. 1).

The MRI scanner relies on an extremely high, uniform, stationary (mono-axial) background field ($B_0$) - the actuating field can never move or change, it is always large (in our case, 7T) and in the global Z direction. $B_0$ is so large that no magnetization can be stored in any material in the bore of the scanner. This means that, for example, the magnetic signature of an NdFeB component would be instantly overwritten by $B_0$ leaving zero torque. We harness the easy axis of magnetization to generate aligning torque without relying on embedded magnetization. The easy axis of magnetization has been harnessed before for actuation purposes at the microscopic scale[26] but never in the MRI bore and never at the meso-scale. This design offers a solution for a soft, miniaturizable, shape-forming catheter which can be actuated and sensed, and therefore controlled, via the MR system. With this proof of concept we demonstrate the design principle, modeling and operation of the CoSMA. Even at the present scale, the increased dexterity and reduced material stiffness of our design can offer advantages over the current state of the art. Nevertheless, further miniaturization of the design will both increase reachable space and reduce signal void size.

## Results

### Free Space Demonstration

We show a sample of the range of motion of the CoSMA in free space in Fig. 2 and in Supplementary Video S2. Sleeve C, with a referential ring angle $\theta_{C0} = 85°$ converges upon, but can never exceed 85° of deformation (Fig. 2G). This curvature can be modulated at run-time by inserting a nitinol stiffening rod down the central channel (Fig. 2D). Rotation of sleeve C permits the full 360° range of motion. Sleeve B, with a referential ring angle $\theta_{B0} = 45°$ is shown in the forward facing, lowest elastic energy, configuration (Fig. 2C). The sigmoidal form (Fig. 2B) is achieved after a 180° rotation of sleeve B when sleeve B is above the threshold length (see Methods Section). The backward-facing configuration (Fig. 2F) is demonstrated after a 180° rotation of sleeve B when sleeve B is below the threshold length (see Methods Section). All of these deformations are attainable in three dimensions as sampled in Fig. 2H–J.

### Clinical Applicability

To demonstrate the potential for clinical applicability of the CoSMA we navigate the various bifurcations of a 3D printed aortic arch phantom (https://www.printables.com/model/661477-aorta). Navigating the aortic arch is essential in procedures such as angioplasty, endovascular aneurysm repair, and cardiac catheter ablation. Current clinical practice for cardiac catheterization involves the manual insertion of a relatively stiff (E ≈ 200 MPa[27], see Supplementary Method S2) pre-bent guidewire from the radial (upper limb) or femoral (lower limb) arteries followed by navigation to the aortic arch. The aortic arch gives rise to four main branches: the right and left common carotid arteries and the right and left subclavian arteries.

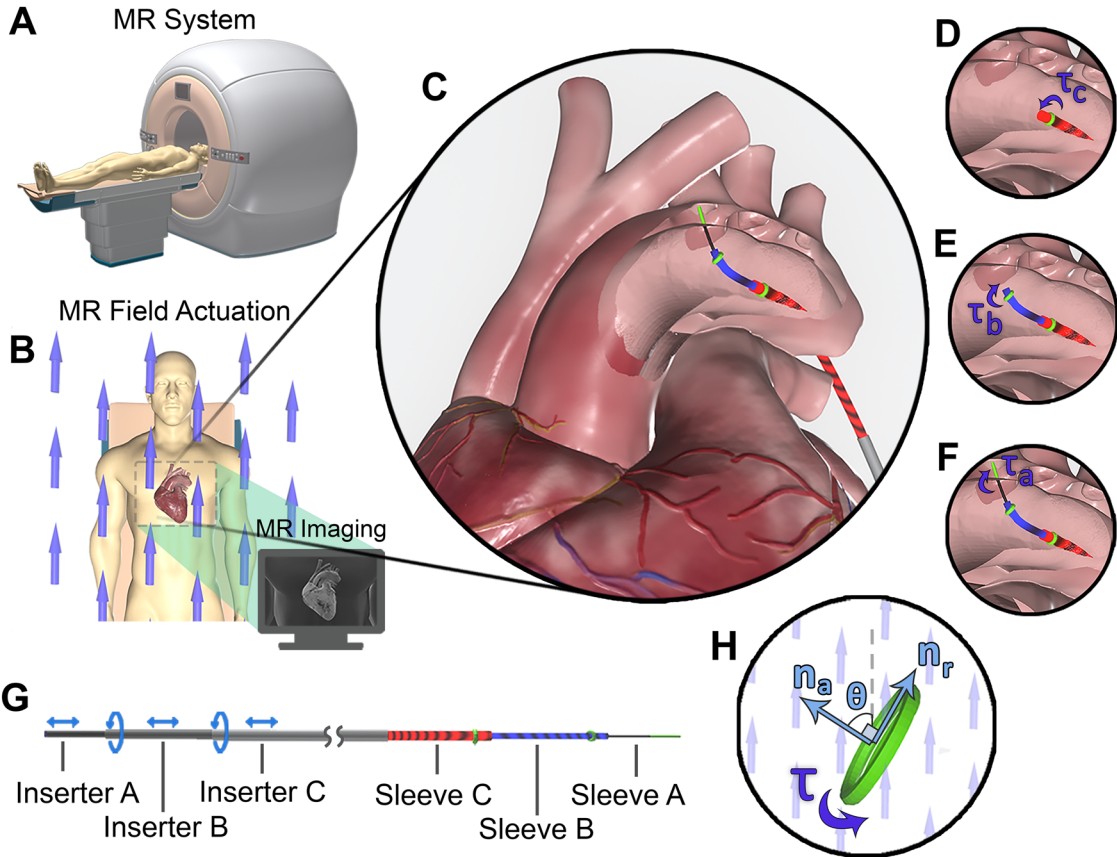

**Fig. 1 | The operating principle of our Coaxial Sleeve Magnetic Actuator (CoSMA).** All MRI systems provide a very strong, static homogeneous background field - $B_0$ (**A**, **B**). Softly ferromagnetic rings align their path of lowest demagnetization ($n_r$) with $B_0$ (**H**). These rings are connected to a series of coaxial sleeves (**C–F**), translation and orientation of which is controlled by the operator (**G**) - these base configurations map to deformations of the CoSMA (See Supplementary Video S1). The entire system can be monitored in high definition 3D via MR Imaging.

**Fig. 2 | The CoSMA deforming in the bore of the MRI scanner (see Supplementary Video S2) and the results of0 the simulation described in Methods Section.** Sleeve C is shown in red, sleeve B in blue and sleeve A in pink with the magnetically active components shown in green. Sleeve C, with a referential ring angle $\theta_{C0} = 85°$ converges upon, but cannot exceed 85° of deformation as shown in (**G**). Sleeve B, with a referential ring angle $\theta_{B0} = 45°$ can be easily rotated to alternate between the sigmoidal deformation mode in (**A,B**) and the forward facing, lowest elastic energy, configuration in (**C**). Sleeve A will always align with $B_0$ which, up to this point, is forward. If the 180° rotation of sleeve B is performed below the threshold length of sleeve B as in (**E**) (see Methods Section for details on this transformation) the primary backward-facing configuration is achieved as in (**F**). At this point sleeve A will align with $B_0$ in the backward-facing orientation. For required deformations of sleeve C below 85° a nitinol stiffening rod can be inserted shown in (**D**). **H–J** show three examples of the three-dimensional shapes available by simply rotating sleeves B and C relative to each other and to the global frame. **K** Shows the three-dimensional point cloud of possible sleeve B and C tip positions generated from the simulation in Methods Section with four sample catheter shapes.

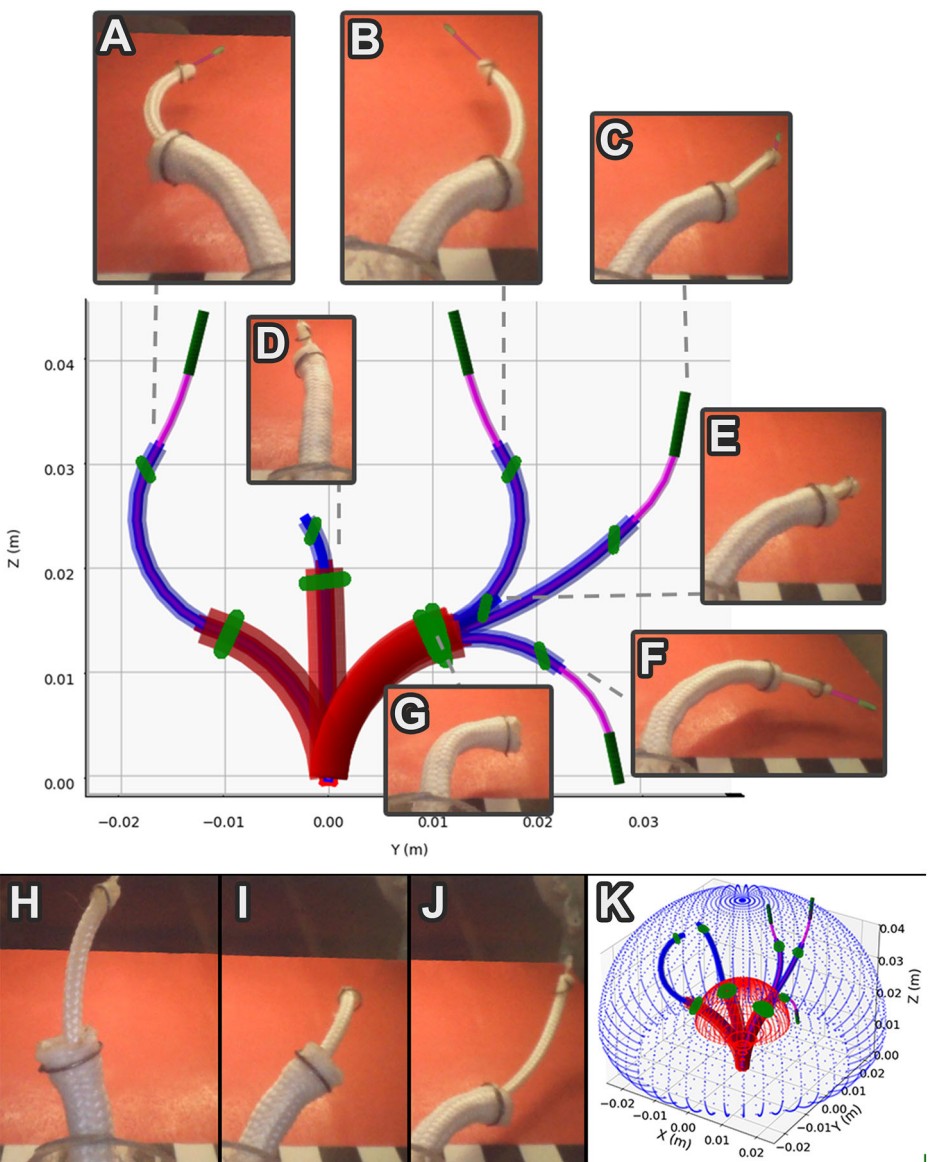

Cardiac catheterization is one of the most common cardiac procedures with more than 1,000,000 performed, either diagnostically or therapeutically, in the United States annually[28]. Intra-procedural complications have been reported with occurrence rates up to 6% and post-procedural complications with occurrence rates up to 33%[29]. Procedural success is reliant, amongst other variables, on the technical skills of the operators[28]. Assisted catheterization, as well as offering a reduction in stiffness, promises to remove this dependency and increase procedural precision[30,31].

The demonstration in Figs. 3, 4 and the Supplementary Video S3 illustrates the potential for a trans-femoral approach in which the CoSMA is guided into any branch of the aortic arch, enabling compatibility with the aforementioned procedures, whilst highlighting the CoSMA's dexterity and trackability under MR guidance. Both Figures are divided into (A) optical camera images, (B) 2D slice of the MR Image (all MRIs were taken in gadolinium doped water (0.5mM Gd-solution)), (C) Reconstruction of anatomy and CoSMA shape based on 3D MRI data (in 3D Slicer software) and (D) 3D rendering of phantom and reconstructed CoSMA shape with signal void centers rendered as gray spheres; camera positions are also shown. The 3D aortic arch prior to CoSMA insertion is shown, then the positions of the CoSMA after navigation into the left subclavian artery

(LSA), the left common carotid artery (LCA), the ascending aorta (AA), the right common carotid artery (RCA) and finally, the right subclavian artery (RSA).

All navigations were performed three times using the prototype catheter under visual feedback via manual operation of the base configuration as detailed in Supplementary Method S5. The LSA (mean navigation time: 62 ± 24 s), LCA (147 ± 81 s) and RCA (155 ± 61 s) all require forward facing sigmoidal deformations of varying radii of curvature, these being the simplest to navigate. The AA (139 ± 15 s) required retroflexion to navigate, and the most demanding route, the RSA (188 ± 45 s), required three-dimensional sigmoidal deformation. Nevertheless, a completely inexperienced user managed to navigate to all five targets in under ten minutes. These navigations were performed in air for the purposes of timings and recording the Supplementary Video S3. MR images are distorted by the presence of ferrous bodies in the MRI bore; this distortion, along with imaging time, has been experimentally minimized (Supplementary Method S4). The full navigations were repeated in gadolinium doped water (0.5mM Gd-solution) to obtain the corresponding 3D MR scans (See Supplementary Method S4 for details) taken at the final pose.

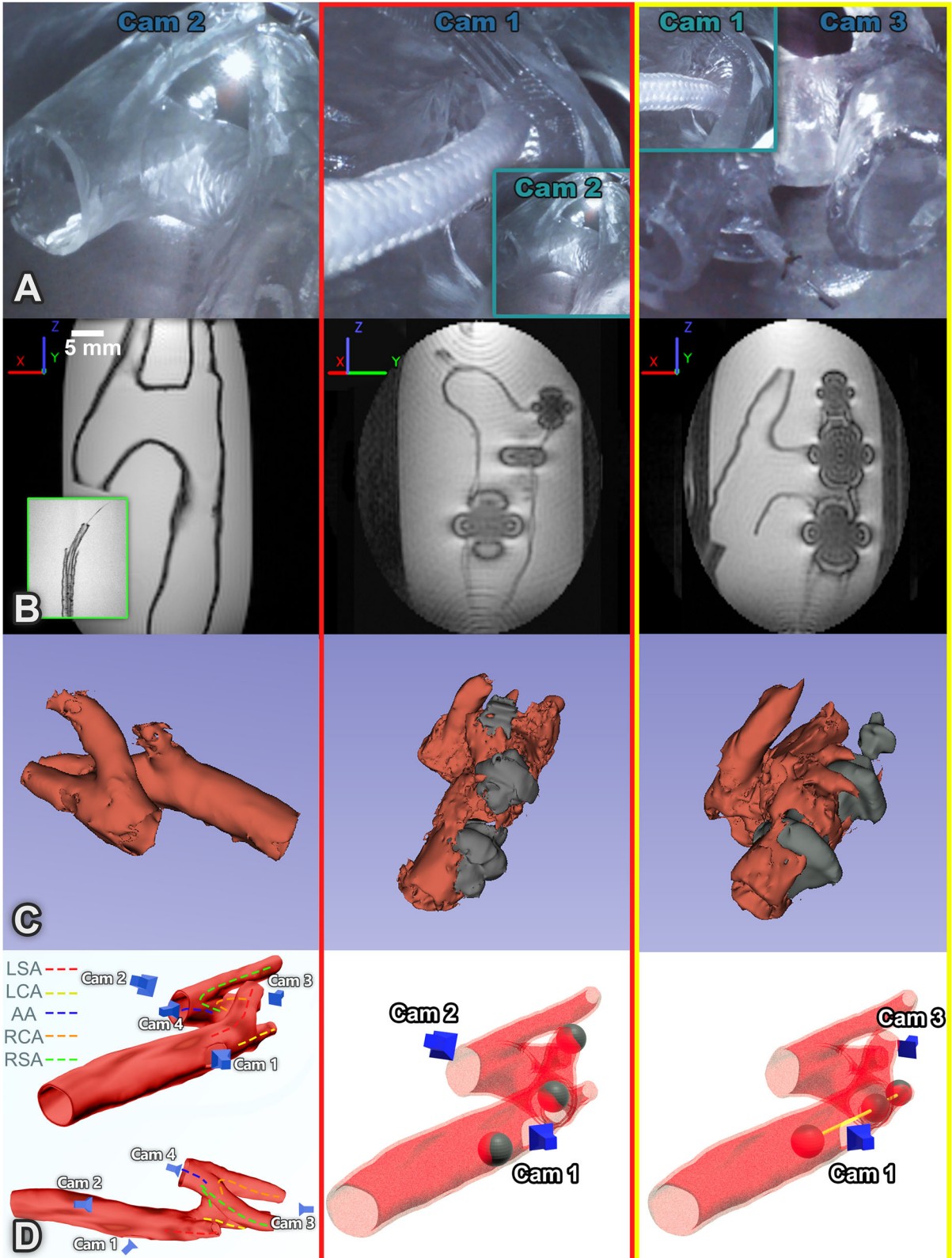

**Fig. 3 | Visual and MR images of the aortic arch navigations.** The left-hand column shows the 3D aortic arch prior to CoSMA insertion. The full shape of the phantom can be seen in row C, reconstructed from the 3D MRI. MR images in row B are selected slices so don't show the full geometry. The central column (red surround) shows the CoSMA navigating into the left subclavian artery (LSA). The right-hand column (yellow surround) shows navigation into the left common carotid artery (LCA). **A** Optical camera images. **B** 2D slice of the MR image showing signal voids from ferrous components. Basis frame shown where $B_0$ always exists in the Z direction. Also shown as insert (green surround) is an MRI of the three-sleeve CoSMA with all metallic components removed. **C** Reconstruction of anatomy and CoSMA shape based on 3D MRI data (3D Slicer 5.8.1 https://www.slicer.org/), and **D** 3D rendering of phantom with reconstructed shape of the CoSMA, signal void positions (shown as gray spheres) and camera positions. All navigations are shown in full in Supplementary Video S3.

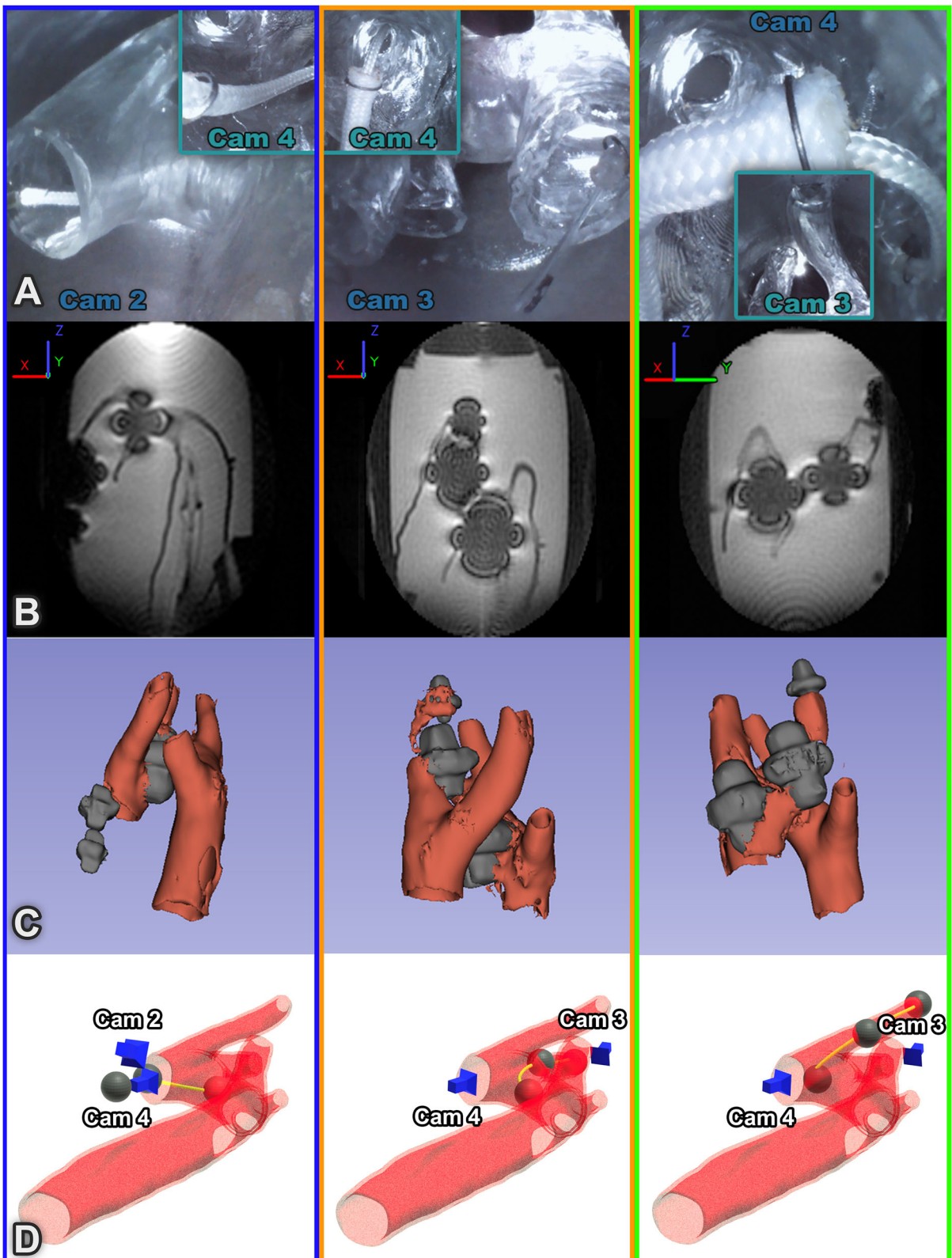

**Fig. 4 | Visual and MR images of the aortic arch navigations (continued).** The left-hand column (blue surround) shows the ascending aorta (AA). The central column (orange surround) shows the right common carotid artery (RCA). Finally, the right-hand column (green surround) shows navigation into the right subclavian artery (RSA). **A** optical camera images, **B** 2D slice of the MR Image showing signal voids from ferrous components. Basis frame shown where $B_0$ always exists in the Z direction. **C** Reconstruction of anatomy and CoSMA shape based on 3D MRI data (3D Slicer), and **D** 3D rendering of phantom with reconstructed shape of the CoSMA and camera positions.

## Conclusion and Discussion

Our catheter design is capable of large multi-DoF deformations which are mechanically controllable from outside the MRI system. Evidently, CoSMA's body is significantly softer than would be possible without harnessing the magneto-static energy of the system. Furthermore, we have used anisotropic material composition (braided nylon) allowing maximization of the twisting to bending stiffness ratio (GJ/EI - See Supplementary Method S2). This enhanced ratio reduces the tendency of the device to exhibit the snap-through instability[5]. We have demonstrated a potential clinical use case with manual navigation under visual feedback into all the branches of the aortic arch.

The geometry and referential orientation of the rings is critical to performance and we have shown how this design feature is encoded with our FE model and in our multi-sleeve numerical model (Methods Section). The principle benefit of magnetic actuation lies in the potential for miniaturization and the only barrier to significant improvement on our 4 mm maximum diameter design lies in fabrication. MR image distortion has been experimentally minimized (Supplementary Method S4) and clearly image distortion/signal voids would reduce further for a more miniaturized system. An earlier prototype of our catheter design leveraged thermoplastic elastomer (TPE) tubes from Arkema (https://www.arkema.com/global/en/) but the lack of torsional rigidity precluded the transmission of twist along the catheter length. This challenge can be mitigated using reinforced TPE (the only commercially available product we have thus far sourced uses non-MR compatible, stainless steel, reinforcing).

Regarding image distortion, the plumes caused by the magnetic entities would vary in size as a function of $|B_0|$. A smaller background field (e.g. or 3T or 1.5T clinical scanner) would reduce the signal void[32] without reducing magnetic torque, as long as magnetic saturation was still achieved (Equation (1)). The aim of this work was to visualize the CoSMA as accurately as possible rather than to optimize the imaging methods or to focus on scan time reduction. Indeed, no dedicated undersampling or reconstruction techniques, such as parallel imaging, compressed sensing or AI-based acceleration methods, which are available on clinical MR systems, have been employed. The orientation of this plume is always aligned with $B_0$ but the shape of the plume has been observed to vary as a function of the orientation of the ring. This makes navigation exclusively under MRI more challenging as tube orientation cannot be directly observed. There is a future work here on, either, analyzing the shape of the plume to determine the pose of the rings or, the addition of fiducial markers to extract CoSMA pose from the MR image. We have also observed the catheter to be agnostic to the imaging gradients with no measurable deformation observed during the imaging cycle.

Here we have presented a proof of concept for a new class of MR-actuated and imaged concentric tube catheter. This article demonstrates a feasible solution to the significant challenge of developing a multi-DoF shape-forming, miniaturizable, self-sensing soft catheter. Future developments should focus on four major areas.

(1) The rapid harvesting and post-processing of MRI information such that this potential wealth of sensory information can be usefully implemented by an operating model. Optimization of imaging methods to focus on scan time reduction, dedicated undersampling and/or reconstruction techniques such as parallel imaging, compressed sensing and/or AI-based acceleration methods, which are available on clinical MR systems, can been employed. At present the images shown are of three distinct signal voids. For live tracking there will be poses where the voids are superimposed in the same physical space. In some cases a void may be completely obscured by a larger void in the same 3D location. We will, however, know the base configurations of the three sleeves and the poses of the three sleeves at an incrementally earlier time step when we had a cleaner image. From a noisy image, base configuration information coupled with a physics model, and previous pose information (image flow) it should be possible to reconstruct the current pose of the three sleeves.

(2) The operational hardware of the CoSMA (control of the position and orientation of the various sleeves) must be automated to allow actuation in response to feedback. This relies on a 5 DoF, MR compatible motor arrangement. Currently, the system is manually operated based on pre-planned base configurations.

(3) The current numerical model will need to be faster and more robust before it can offer a path to a closed-loop control solution. Controlling such a design comes with its own challenges which have a lot in common with the control challenges of elastic concentric tubes[33,34] but also rely on formulation of the magnetic energy and torque[35]. In this article we make no attempt to perform formal control, the catheter is manually actuated, and all the modeling is performed offline, prior to navigation. As this is a proof of concept of the design, the formal controllability of the catheter remains unaddressed but from our inquiries it appears this will be possible through the well encoded channels presented in the early concentric tube publications. Indeed, in terms of stability (a fundamental requirement for controllability) our design offers significant improvement over traditional elastic concentric tubes due to the vastly improved ratio of GJ/EI (Supplementary Method S2.2), a well documented limiting factor regarding the snap-through instability. In terms of a controllability comparison to traditional magnetic actuation, for tip-driven/axially magnetized catheters this is effectively a solved problem (e.g., ref. 36). For multi-DoF (active shape forming) catheters such as this, the principle of how we could control our system is almost identical to the theory presented in ref. 37.

With these three key developments, the prospect of autonomous navigation will become realistic.

Finally, fabrication of the CoSMA should be both automated and miniaturized. The manufacturing and positioning of the magnetic elements currently represents a significant source of operational error. Furthermore, delicate anatomy must be protected from the rigid and potentially sharp edges of the metal hoop. Laser cut and burred rings with a thin layer of high stiffness elastomer (e.g. PDMS) can mitigate both of these issues. Our next activity is to miniaturize the design, something which will both increase reachable space and reduce signal void size. The only barrier to the CoSMA having an outer diameter of the order of 1 mm lies in sourcing MR compatible reinforcing materials.

## Methods

### Principle of Operation

Any magnetically actuated soft robot operates via balancing internal elastic energy with an externally applied magnetic field. Primarily, traditional magnetic soft robots have relied on a hard magnetic remanence stored within the robot's structure, actuated by relatively low externally applied fields[38]. Within the bore of any MRI system, we have an ultra-high, mono-axial background field ($B_0 \in (1, 7)T$) and a much smaller tri-axial gradient ($\nabla B_0 \in (0, 600)mTm^{-1}$) available for actuation purposes (the magnitude of the radio-frequency fields are too small to impact actuation). The background field is sufficiently strong that no magnetic remanence can be retained. Thus, the concept of "magnetically hard materials" whereby materials hold a permanent vector of magnetization becomes redundant and, consequently, well-explored modes of magnetic manipulation become obsolete[39].

A non-spherical magnetic object exposed to a background field will experience a torque attempting to align the longest (principle) axis with the background field[26,40]. This easy axis alignment is traditionally considered to be too weak for robotic applications above the micro-scale but a sufficiently large field (such as the $B_0$ field of a clinical MRI system) can be leveraged to generate considerable deformation - orders of magnitude higher than the deformation available from an equivalent volume of metal under gradient actuation. Our design takes advantage of the large magneto-static energy available via the $B_0$ field and moves the actuation of controllable DoFs outside of the system. Magnetic energy is provided by the background field, and ferrous rings embedded at the tip of each sleeve translate this energy into deformation. We cannot directly use the stationary background field ($B_0$) for actuation in any controllable direction, only the provision of actuating energy, we therefore must provide directional actuation through some alternative mechanism. Through rotation and translation of the respective

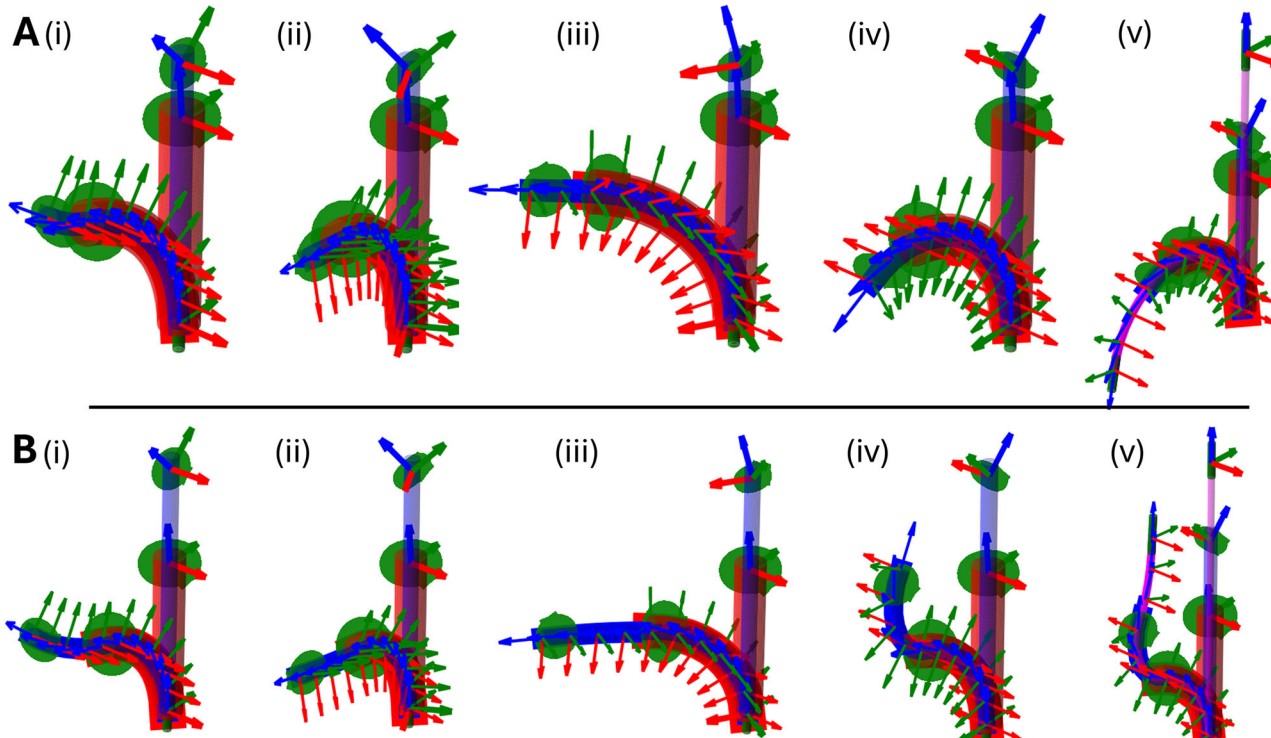

**Fig. 5 | The output of our numerical simulation showing two step-wise progressions.** Referential pose is straight in positive Z (vertically upwards here), sleeve B in blue, sleeve C in red, referential ring positions in green with their basis frames (red-green-blue) in bold. Deformed pose is shown in the same colors with local basis frames along the continuum length. The homogeneous background field is in positive Z. **A** Sleeve B is 20 mm longer than sleeve C (initially sleeve A length = 0). (i) Rotation of sleeve B = 0°, this represents the default pose for this design. (ii, iii, iv) Sleeve B is rotated through 180° in 60° increments (input rotations shown as orange arrows, end effector rotations as purple arrows). Retroflexion is first observed when Sleeve B rotation exceeds 120°. (v) Sleeve A is inserted and, due to the retroflected Sleeve B, aligns negatively with $B_0$. **B** Sleeve B is 30 mm longer than sleeve C Again, sleeve B is rotated from 0° to 180° in 60° increments but the increased length of sleeve B induces a forward facing sigmoidal deformation. In (v) sleeve A is inserted and aligns positively with $B_0$.

sleeves we can create a variety of shapes which ultimately enable tortuous navigation (Fig. 1). Thus, the control inputs become the base configurations of the catheter as opposed to, as with traditional magnetic catheter designs, the actuating field settings. The choice of rings as the magnetically active component (as opposed to ellipsoids or rods) derives from their natural conformity to the cylindrical shape of the CoSMA. Slender, linear rods mounted at various angles along the length of the catheter are easier to model (see Supplementary Method S3) but inherently length-limited and thus torque-limited. These rings are agnostic to imaging gradients and can thus be simultaneously imaged. The design principle is conceptually similar to that of the concentric tube designs in refs. 41,42 (which also rely on rotating and translating cores) but, as deformation energy is provided magnetically rather than via elastic restoration, the material construction of the CoSMA can be of the order of three orders of magnitude softer. This relative softness can improve patient safety but, less intuitively, when combined with the material anisotropy of the braided design (See Supplementary Method S2) also mitigates the snap-through instability commonly observed in concentric tube designs[5].

Magnetic torque due to easy-axis alignment can be shown to be a function of geometry - demagnetization factors ($n_r$, $n_a$) and volume ($v$), saturation magnetization of the material ($m_s$), and $\theta$, the angle between $B_0$ and the global frame direction of $n_a$[35] (Fig. 1G, see Supplementary Method S3) for a detailed explanation),

$$\tau_{mag} = \frac{1}{2}\mu_0 v |n_r - n_a| m_s^2 sin(2\theta), \qquad (1)$$

where $\mu_0$ is the vacuum permeability.

Due to the presence of $2\theta$ in the sine curve, the torque decays to zero every 90° ($\theta$ is the angle between the axis of symmetry of the magnetic body

and the (stationary) background field - Fig. 1). This is physically evident when we consider that as we rotate a slender object in $B_0$ (Fig. 1H) the induced magnetization will flip every 180°, i.e., every time $\theta$ passes through zero. This stands in contrast to the full 360° range of motion available when magnetically remnant bodies are placed in relatively low actuating fields (e.g.[10]). Due to this significantly different operating principle, any single magnetic body can only generate catheter deformation up to but not exceeding 90°. As such, we have employed the concentric tube approach to embed a second magnetic body with a distinct geometric arrangement (varying referential alignment). Via rotation and translation of the two bodies relative to each other we demonstrate how it is possible to generate catheter deformation in excess of 90° despite this fundamental limitation of the actuating platform (the MRI scanner). A near-orthogonally mounted ring (e.g. referential angle, $\theta_0 = 85°$, see Fig. 2) will, depending on magnetic energy and elastic stiffness, asymptotically converge on an 85° deformation. This curvature can be regulated in the design phase via magnetic geometry and in the actuation phase via stiffness modulation. Thus, our outer sleeve (henceforth referred to as sleeve C), manufactured from 3 mm diameter nylon braid (see Supplementary Method S6 and Supplementary Video S4) and with a 4 mm diameter iron ring affixed to the tip, can provide deformation up to, but never exceeding, $\theta_{C0} = 85°$ (in the numerical formulation these angles and torques are three-dimensional but for the purposes of this conceptual explanation we adopt planar notation).

A second sleeve (sleeve B), manufactured from 1.5 mm diameter nylon braid, is free to translate and rotate within sleeve C but constrained to follow the same curvature up to the distal point of sleeve C. Were sleeve B to host a ring with the same referential angle as sleeve C it would be subject to the same constraints and thus, deformation beyond 90° (retroflexion) would not be possible. The referential ring orientation on sleeve B ($\theta_{B0}$) is one of the key variables considered in the numerical formulation (Methods Section).

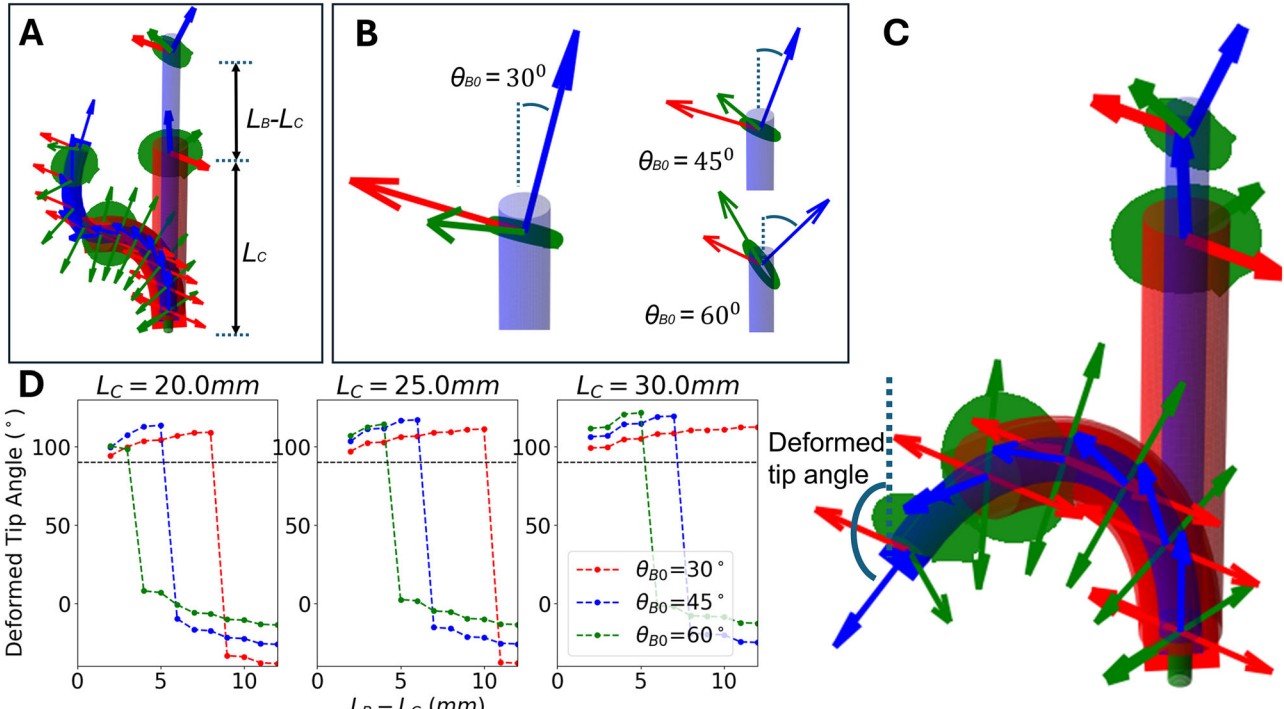

**Fig. 6 | Curves showing the unconstrained length of sleeve B versus the deformed tip angle of sleeve B.** Specifically, **A** illustrates the parameters for the length of sleeve C ($L_C$) and the unconstrained length of sleeve B ($L_B - L_C$). In this particular sample the deformed configuration is sigmoidal (deformed tip angle < 0˚). **B** Shows three different referential angles of the ring at the tip of sleeve B ($\theta_{B0} = 30$˚, 45˚, 60˚) - this is the angle between the symmetrical axis of the ring and the applied field. **C** Shows the deformed tip angle parameter as well as a sample configuration where the deformed tip angle > 90˚ - an example of the retroflected pose. **D** Plots the unconstrained length of sleeve B (X-axis) against deformed tip angles (Y-axis) for the three referential angles of ring B - the black checked line represents the criterion for retroflexion. As can be seen, smaller $\theta_{B0}$ (red curve) gives a larger stable region of retroflexion - i.e. retroflexion is possible for longer lengths of unconstrained sleeve B - but struggles to produce sigmoidal deformations (plot on the right). Conversely, larger $\theta_{B0}$ (green curve) readily produces the sigmoidal deformation but struggles to retroflect (plot on the left). For these settings, $\theta_{B0} = 45$˚ gives a stable blend of retroflexion and sigmoidal deformations. A more exhaustive parameter optimization lies beyond the scope of this work, but this analysis justifies our design decision pertaining to $\theta_{B0}$.

Clearly, the magnitude of the magnetic torque relative to that on sleeve C is critical but, assuming a correctly balanced system, retroflexion can be achieved via manipulation of $\theta_{B0}$. Letting $\theta_{B0} = 45°$, sleeve B will either align in the forward facing, lowest elastic energy, configuration (Fig. 2C) or, via a 180° rotation of sleeve B, will align in the backward facing configuration (Fig. 2F).

A third and final sleeve (sleeve A), manufactured from 0.4 mm nylon wire, carries a straight iron pin which is referentially aligned with $B_0$, i.e. $\theta_{A0} = 0°$. Clearly, this pin displays different demagnetization factors to the rings leveraged in sleeves B and C, but aside from this, the principle of operation is identical. Sleeve A can be inserted beyond sleeve B or fully retracted out of the system as required. The purpose of this sleeve is simply to extend the range of motion of the CoSMA. Importantly, dependent on the configuration of sleeve B, sleeve A will align either forward (Fig. 2A) or backward facing (Fig. 2F).

We use annealed iron wire to fabricate our ferrous components due to high saturation magnetization ($m_s = 1.43 \times 10^6 (Am^{-1})$[43]). Determination of the demagnetization factors is a non-trivial exercise covered in Supplementary Method S3. Clearly, the angle between external field and symmetrical axis ($\theta$) is a function of both referential pose and catheter deformation and so a simulation is developed which employs a pseudo-rigid link model to balance elastic and magnetic torques throughout the coaxially connected sleeves. We use braided nylon sleeves for our continuum structure as we must be able to transfer torsion whilst allowing bending. The mechanical behavior of these sleeves is characterized in Supplementary Method S2.

**Simulation**

Having characterized the mechanical properties of the continuous structures from which the CoSMA is constructed (Supplementary Method S2) and the magnetic behavior of our ferrous components (Supplementary Method S3) we have sufficient data to populate a rigid link model (according to Supplementary Method S1) of the complete system.

Each sleeve was discretized into n pseudo rigid links with joint angles $Q \in \mathbb{R}^{n \times 3}$. A point torque is applied at the tip determined according to Supplementary Method S3 and the orientation of the magnetic element, itself a function of the joint angles, $\tau_{mag}(Q) \in \mathbb{R}^3$. Notably, $\tau_{mag}$ from equation (1) is planar (magnitude only) so, to convert this to $\mathbb{R}^3$, the torque direction must be ascertained. This is the unit vector of the cross product of the background field with the symmetrical axis of the magnetic element ($\tau_{mag} = |\tau_{mag}|(\widehat{B \times r})$). Elastic torques ($\tau_{ela}(Q) \in \mathbb{R}^{n \times 3}$), determined from the stiffness matrices in Supplementary Method S2 are balanced with magnetic torques according to $J\tau_{mag} = Kq$ where $J$ is the robot Jacobian as described in Supplementary Method S1 of each of the respective sleeves. Hence, there are different $J$, $K$ and $\tau_{mag}$ values for each of the three sleeves. This torque balance propagates along each individual sleeve producing independent deformations ($Q_A$, $Q_B$, $Q_C$).

The position, but not orientation, of each sleeve must be equal up to the distal point of each sequential sleeve. For instance, sleeve B is free to rotate but must be in the same location as sleeve C (as it is concentric) up until the end of sleeve C. Beyond this point sleeve B is unconstrained. To encode this constraint, an interactive force is applied equally and oppositely at each joint along the length of both concentric sleeves. This interactive force is computed according to the stiffness matrix and the vector difference in position at each location (e.g., $p_{Bi} - p_{Ci}$). This constrains (after some iterative loops) the concentric sleeves into the same position, despite having independent lengthwise orientation ($q_z$ in the robot frame in Fig. 5) and therefore not necessarily having the same 3D joint angles. As a consequence of this modeling feature, the lengthwise orientation of each sleeve (roll) can be independently controlled as an input. This iterative numerical solution was

solved (for $n = 24$ in the results presented), using an actively damped ($\mu(err) \in (2\%, 25\%)$ calculated as a function of current error) Newton-Raphson method converging in the range of 0.5–10 s.

The principle design feature of the CoSMA is that of retroflexion. The referential orientation of ring C ($\theta_{C0}$) is as close to 0° as possible (The symmetrical axis of the ring is aligned with the background field) meaning sleeve C deforms to an angle asymptotically approaching 90°. Sleeve B can then rotate and/or translate to achieve a wide variety of shapes, for some of which the tip of sleeve B is deformed beyond 90° (Fig. 5 and Fig. 2). This allows sleeve A to align either positively or negatively with the background field permitting the full 360° range of motion demonstrated in Fig. 2.

It is self evident that the torque magnitude of each sequential sleeve must be lower than the previous so as to not overpower the shape to which it is conforming, thus $|\tau_A| < |\tau_B| < |\tau_C|$. For any manipulator featuring multiple, linearly dependent, active segments, when one sleeve is translated or rotated, the change in torque affects the pose of the other sleeves which in turn affects the magnetic torque of both. This is a conceptually similar problem to the model we developed in ref. 44 for remnant shape-forming catheters. Sleeve C will always move in response to movement of sleeve B. It is therefore theoretically necessary to move both tubes to a different equilibrium state whereby the pose of tube C remains the same. In practice, for this particular design, as can be observed in supplementary Video S2, the inbuilt imbalance of magnetic torques means sleeve C is almost perfectly stationary as sleeve B rotates and translates. Based on considerations in Supplementary Method S4 regarding image distortion, the volume of metal throughout the CoSMA must be minimized to permit the clearest possible view of CoSMA shape and surrounding anatomy (Fig. S6). For this motive we used the minimum size of one loop for both ring B and ring C and the pin on sleeve A is 5 mm long. This arrangement is the smallest metal volume which provides sufficient torque to provide a full range of motion (for this particular set of sleeve materials and geometries). The constraint $\tau_B < \tau_C$ is met under this arrangement as the diameter of ring B is half that of ring C (see Table S2).

To ascertain a suitable angle to position ring B onto sleeve B we performed the numerical optimization shown in Fig. 6. The unconstrained length of sleeve B (defined as the length of sleeve B minus the length of sleeve C) is plotted against the deformed tip angle of sleeve B after the 180° rotation of sleeve B (Fig. 5Aiv, Biv). Any deformed tip angle in excess of 90° represents retroflexion. For a longer unconstrained sleeve B, the CoSMA will deform into the sigmoidal shape as sleeve B rotates (Fig. 5Biv). This shape is represented in Fig. 6D as the right hand region of negative tip angles. The angle at which ring B is mounted (represented by the different curves) determines the length of sleeve B which can stably support retroflexion before the CoSMA flips into the sigmoidal deformation mode. It can be observed that smaller $\theta_{B0}$ (red curve) gives a larger stable region of retroflexion - i.e. retroflexion is possible for longer lengths of unconstrained sleeve B - but struggles to produce sigmoidal deformations. Conversely, larger $\theta_{B0}$ (green curve) readily produces the sigmoidal deformation but struggles to retroflect. For these settings, $\theta_{B0} = 45°$ gives a stable blend of retroflexion and sigmoidal deformations.

## Data availability

Any additional information required to reanalyze the data reported in this paper is available from the lead contact upon request.

## Code availability

Any code utilized in this paper is available from the lead contact upon request.

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

## Acknowledgements

This work was supported in part by the Engineering and Physical Sciences Research Council (EPSRC) under Grants EP/Y037235/1 and EP/V009818/1, the European Research Council (ERC) through the European Union's Horizon 2020 Research and Innovation Program under Grant 818045, and by the National Institute for Health and Care Research (NIHR) Leeds Biomedical Research Center (BRC) (NIHR203331). Additionally, JES acknowledges funding from the Wellcome Trust (219420/Z/19/Z) and the MRC (MR/X502789/1). Any opinions, findings, and conclusions, or recommendations expressed in this article are those of the authors and do not necessarily reflect the views of the EPSRC, the ERC, the NIHR, the Wellcome Trust, or the MRC. We also thank Leah Khazin, Ampere Kui and Samwise Wilson for technical support throughout. Peter Lloyd and Nikita Murasovs contributed equally to this work and should be considered joint first authors. J. Chandler was supported by the Leverhulme Trust and the Royal Academy of Engineering under a RAEng/Leverhulme Trust Research Fellowship under grant LTRF-2425-21-154.

## Author contributions

P.L. and N.M. conceived, designed and executed the study. P.L., N.M., and Y.L.M. designed and constructed prototypes. J.E.S performed MRI acquisition. Y.L.M., A.B., and V.F. contributed to data aggregation and analysis. B.C. and J.D. contributed to results analysis. P.L., J.E.S., and P.V. supervised the project. P.L. and N.M. wrote the manuscript. J.H.C., E.D.A., J.E.S., and P.V. revised and reviewed the manuscript. All authors discussed the results and commented on the manuscript.

## Competing interests

The authors declare no competing interests.
