## [Transparent Peer Review file · Communications Engineering]

A Concentric Tube Catheter for Endoluminal Interventions, Steered and Imaged via Magnetic Resonance Imaging

Corresponding Author: Dr Peter Lloyd

Version 0:

Reviewer comments:

Reviewer #1

(Remarks to the Author)

This paper presents a structurally adaptable "Coaxial Sleeve Magnetic Actuator" (CoSMA), driven by the background field of the MRI scanner. The authors presented the concept design, magnetic control method, and experiments under the MR scanning. However, there are similar concepts of concentric tube-like magnetic catheter in the recent works. The current version hasn't elaborated enough on the uniqueness of the proposed design and the necessity of such a design under the MR environment. The quality of the figures should be improved. Please see the detailed comments below:

1. The concentric tube magnetic continuum robot has been published in recent works; it is recommended that the authors make specific comparisons in terms of the structure design, working space, tracking method, and actuation mode of the magnetic field, for example:

-M. Richter, et al., "Concentric Tube-Inspired Magnetic Reconfiguration of Variable Stiffness Catheters for Needle Guidance," in *IEEE Robotics and Automation Letters*, vol. 8, no. 10, pp. 6555-6562, Oct. 2023, doi: 10.1109/LRA.2023.3307294

-Z. Li, et al., "Design and Hierarchical Control of a Homocentric Variable-Stiffness Magnetic Catheter for Multiarm Robotic Ultrasound-Assisted Coronary Intervention," in *IEEE Transactions on Robotics*, vol. 40, pp. 2306-2326, 2024, doi: 10.1109/TRO.2024.3378442.

And what is the core advantage of the proposed design over these works?

2. The authors should further elaborate on why the MR-driven catheter should be designed as a concentric tube shape. What is the advantage of this shape, considering the MR environment?

3. It's claimed by the authors that the outer diameter of the proposed concentric tube catheter is 3 mm, 1.5 mm, and 0.4 mm. However, in the supplementary video, the size of the catheter seems relatively large, very different from the conventional magnetic catheter. Can the authors supplement the measurement figures of the catheter by using measurement tools? It's doubtful that such a large size can navigate through small vessels. Please explain this.

4. The MR magnetic field is very strong, and it is very difficult to control the field to actuate a robot inside. The authors should elaborate on the control method of the MR magnetic field to ensure the controllability of the magnetic catheter in such a strong field. How is the control accuracy of the magnetic catheter using the MR magnetic field compared to the normal permanent/electromagnetic field?

5. How can the magnetic field actuate the desired tube while keeping the other tube stationary to ensure the accuracy of the distal tube, as they are all affected by the magnetic field? This is a very critical issue for a magnetic catheter integrated with multiple magnetic components.

6. One major motivation for an MR-compatible robot is the utilization of MR images as the closed-loop feedback to form a closed-loop controller. Have the authors considered this issue during the design process?

7. It's recommended that the authors should revise the abstract, removing (a) and (b), and replace them with "first" and "second" instead.

8. It's recommended that the author should give the full name of abbreviations when first using them, e.g., MRI—"Magnetic Resonance Imaging". The authors should use the term "MR" for many locations throughout the paper instead of "MRI", e.g.,

in Figure 1, the MRI field should be considered revised to "MR Field".

9. The figures should be further improved and edited, considering the standard of the current publications, e.g., <https://doi.org/10.1038/s44172-025-00424-3>

10. In Supplementary Video S4, the fabrication process illustration is shown. However, it is recommended that the authors supplement the real picture of the proposed catheter in the video at every fabrication stage.

Reviewer #2

(Remarks to the Author)

This study presents a structurally adaptable CoSMA, constructed of two sleeves and one guidewire, driven by the background field of the MRI scanner. It enables multi-degrees of freedom motion. The mathematical simulation and phantom test were performed. However, there are several points that need to be addressed in the calculation and discussion. First of all, this design enables multi-degree of freedom, but it is difficult to operate for a radiologist. The magnetic field is perpendicular to the central axis of the sleeves. The radiologist needs to consider when should the second sleeve be placed and even the third tip. Second, if the operation was wrong, and I think it can be easily wrong, can it damage the surrounding vessel? The worst situation needs to be considered. Third, the distortion or image voids need to be presented when all three components were placed out. I am afraid that the image voids are bigger than the diameter of big vessels around the heart. The size of such a big tip limits the application in small vessels. Finally, it is difficult to localize or track such tips with three components using MRI technique.

Reviewer #3

(Remarks to the Author)

This work introduces a concept of MRI-guided concentric tube catheter based on concentrically actuated sleeves with iron rings wrapped around their outer surface to produce magnetic torques to bend the sleeves based on the interaction between the iron rings and the principal field of the MRI. Although the demonstration provided here is preliminary and still far from clinical translation, the concept is new and worth exploring. This submission provides a solid grounding for further exploring this concept in the future.

Despite this promising proposition, some aspects of the work remain unclear, and I have some concerns regarding the feasibility of this concept in more realistic settings which I detail below.

Comments:

- The experimental demonstration was performed in a preclinical 7-T MRI machine. It is not clear whether the simulated scenarios are also restricted to this principal field value. As this property influences both the artifacts, acquisition time, and dexterity of the device, I think it is important to discuss (at least in simulation/theory) how the proposed design translates to clinical systems (0.5 – 3T).
- I appreciate the efforts of the authors to comment on the artifacts and discuss the value of the Feret diameter. That being said, it is clear that the use of the iron ring presents a fundamental limitation in terms of acquisition time and image quality and artifacts. While these limitations are acknowledged, the authors do not really discuss how these limitations could be overcome in practice to make this concept clinically realistic (i.e. to ensure acquisition time in the second rather than the minute range, and whether the artifacts are expected to be reduced significantly enough to make them acceptable for the target procedure).
- Thermal effects are not discussed: can the authors elaborate on what is the expected heat and possible temperature elevation in the iron rings produced by the RF pulses?
- How are the sleeves insertion/retraction and roll actuated? The device exhibits 5 DOF, so I can see that being a challenge to design an actuation unit that is both compact enough and that does not deteriorate the image quality. It is said the "navigations were performed three times using the prototype catheter under visual feedback via manual operation of the base configuration"; does that mean there was no motorized unit throughout the experiments?

Version 1:

Reviewer comments:

Reviewer #1

(Remarks to the Author)

This paper presents a structurally adaptable "Coaxial Sleeve Magnetic Actuator" (CoSMA), driven by the background field of the MRI scanner. The authors have addressed most of my comments in the revised manuscript. However, there are still several concerns before considering publications. Please see the detailed comments below:

1. The authors should consider revising the title of the paper, as the current title does not reflect the characteristics of the proposed robot. The targeted surgery could also be included in the title to enhance the functionality of the proposed robot.
2. It is recommended that the authors provide a control block diagram including the input/output, MR field control, and MR imaging feedback in the Method section.

3.The authors should give a clear description of the design of the proposed magnetic catheter using a figure including the advancement mechanism. The outer diameter of each tube should be clearly labeled.

4.The figures should be further improved and reedited to be consistent with the journal style.

Reviewer #2

(Remarks to the Author)

The authors have fully addressed my comments. I have no further suggestions.

Reviewer #3

(Remarks to the Author)

The authors addressed all my comments and I have no further remarks or suggestions.

An MRI Actuated and Imaged Concentric Tube Catheter - **Authors Response**

Peter Lloyd and Nikita Murasovs, Yael L. May,
Alistair Bacchetti, Benjamin Calme, Joshua Davy,
Vittorio Francescon, James H. Chandler, Erica Dall'Armellina,
Jurgen E. Schneider, Pietro Valdastri.

Manuscript ID: COMMS-ENG-25-0803-T

Introduction

The authors wish to thank the Editors and Reviewers for their constructive feedback on the paper form and content. We strongly believe the feedback has enhanced the value of the submission and, in the following, provide a justification to any changes in the paper derived from the comments. All comments were considered by the authors in a point-by-point fashion and any changes in the new submission appear as **red text**. In the present response we report comments from the Editor and Reviewers in **green text**, the response from the Authors with **blue text**, and changes made to the original paper with **red text**. Red text extracts are verbatim recreations from the revised paper, as such, wherever figures and citations appear in red, the numbering system pertains to the revised submission, not to this document.

1 Reviewer 1 Comments:

This paper presents a structurally adaptable “Coaxial Sleeve Magnetic Actuator” (CoSMA), driven by the background field of the MRI scanner. The authors presented the concept design, magnetic control method, and experiments under the MR scanning. However, there are similar concepts of concentric tube-like magnetic catheter in the recent works. The current version hasn't elaborated enough on the uniqueness of the proposed design and the necessity of such a design under the MR environment. The quality of the figures should be improved. Please see the detailed comments below.

We thank the reviewer for their input. They have raised some pertinent queries regarding the robotics elements of the design, the literature review and some presentational issues. We have tried to respond in a manner which we hope

they will consider satisfactory.

1. The concentric tube magnetic continuum robot has been published in recent works; it is recommended that the authors make specific comparisons in terms of the structure design, working space, tracking method, and actuation mode of the magnetic field, for example: “Concentric Tube-Inspired Magnetic Reconfiguration of Variable Stiffness Catheters for Needle Guidance”[1], and “Hierarchical Control of a Homocentric Variable-Stiffness Magnetic Catheter for Multiarm Robotic Ultrasound-Assisted Coronary Intervention”[2], And what is the core advantage of the proposed design over these works?

The MRI scanner, uniquely, offers non-invasive, radiation- and exogenous contrast-free high definition 3D imaging of living tissue. It does however rely on a strong, uniform, static background field (B_0). Our vision is to develop a catheter which is actuated via this background field whilst simultaneously imaged, thus allowing intervention and observation in the same procedure. One of the challenges associated with this is the fact that the actuating field can never move or change, it is always large (in our case, 7 Tesla) and in the global Z direction. One consequence of this is that B_0 is so large that no magnetization can be stored in any material in the bore of the scanner, this means that, for example, the magnetic signature of an NdFeB component (commonly used in the literature and used in the papers the reviewer has highlighted) would be instantly overwritten by B_0 leaving zero torque. Our solution to this challenge was to harness the easy axis of magnetization to generate aligning torque without relying on embedded magnetization. The easy axis of magnetization has been harnessed before for actuation purposes (e.g. [3]) at the microscopic scale but never in the MRI bore and never at the meso-scale.

We cannot directly use the stationary background field (B_0) for actuation in any controllable direction, only the provision of actuating energy, we must provide directional actuation through some alternative mechanism. Our solution to this second challenge was to develop the principle of concentric tube robots but using magnetically active rings. We acknowledge that other magnetic concentric tubes have been demonstrated (e.g. [1], [4]) but never before in the context of the uniquely challenging MRI environment. In response to the reviewers comment we have modified our literature review to be clearer about the degree of separation of our design against those existing contributions, including the two citations specifically noted by the reviewer ([1, 2]).

Line 33 of the revised paper: “Concentric tube robots leveraging remnant magnetic actuation to enhance stability [9] [10] or reduce stiffness [11] [12] have been demonstrated but never in an MR compatible format. Here, we offer a fully hybridized approach combining the mechanical principles of concentric tube designs with the off-board energy provision of magnetic actuation to create a softer, and therefore safer, design, MR compatible by design, and with less susceptibility to the snap-through instability.”

Line 60 of the revised paper: “The MRI scanner relies on an extremely high,

uniform, stationary (mono-axial) background field (B_0) - the actuating field can never move or change, it is always large (in our case, 7T) and in the global Z direction. B_0 is so large that no magnetization can be stored in any material in the bore of the scanner. This means that, for example, the magnetic signature of an NdFeB component would be instantly overwritten by B_0 leaving zero torque. We harness the easy axis of magnetization to generate aligning torque without relying on embedded magnetization. The easy axis of magnetization has been harnessed before for actuation purposes at the microscopic scale [3] but never in the MRI bore and never at the meso-scale. This design offers a solution for a soft, miniaturizable, shape-forming catheter which can be actuated and sensed, and therefore controlled, via the MR system.”

2. The authors should further elaborate on why the MR-driven catheter should be designed as a concentric tube shape. What is the advantage of this shape, considering the MR environment?

The principle motive for the concentric tube design is to increase the range of poses available under easy axis of alignment actuation. The key to this lies in Equation 1 in the original article:

$$\tau_{mag} = \frac{1}{2}\mu_0 v |n_r - n_a| m_s^2 \sin(2\theta), \quad (1)$$

and, in particular, the component $\sin(2\theta)$.

Theta is the angle between the axis of symmetry of the magnetic body and the (static) background field. Fig. 1 shows a ring (isometrically projected) being rotated counterclockwise in 45° increments against a background field shown as light blue arrows. The most Northerly point on the ring will always be the North pole and the system wants to align this pole with the direction of \mathbf{B}_0 . Due to the presence of 2θ in the sine curve the torque decays to zero every 90 degrees. This is physically evident when we consider that as we rotate a slender object in B_0 the induced magnetization will flip every 180 degrees, every time θ passes through zero. This stands in contrast to the full 360° range of motion available when magnetically remnant bodies are placed in relatively low actuating fields (e.g. [1]). Due to this significantly different operating principle, any single magnetic body can only generate catheter deformation up to but not exceeding 90° . As such, we have employed the concentric tube approach to embed a second magnetic body with a distinct geometric arrangement (different alignment w.r.t. B_0 under straight/referential catheter configuration). Via rotation and translation of the two bodies relative to each other we demonstrate how it is possible to generate catheter deformation in excess of 90° (i.e. retroflexion) despite this fundamental limitation of the actuating platform (the MRI scanner). This concentric tube design thus introduces 5 DoF actuation which allows for precise control of the robot in 3D and increases available range of motion. We have modified Section 2 'Principle of Operation' to reflect the reviewers query.

Line 114 of the revised paper: “Due to the presence of 2θ in the sine curve,

the torque decays to zero every 90° (θ is the angle between the axis of symmetry of the magnetic body and the (stationary) background field - Fig. 1). This is physically evident when we consider that as we rotate a slender object in B_0 (Fig. 1H) the induced magnetization will flip every 180° , i.e. every time θ passes through zero. This stands in contrast to the full 360° range of motion available when magnetically remnant bodies are placed in relatively low actuating fields (e.g. [10]). Due to this significantly different operating principle, any single magnetic body can only generate catheter deformation up to but not exceeding 90° . As such, we have employed the concentric tube approach to embed a second magnetic body with a distinct geometric arrangement (varying referential alignment). Via rotation and translation of the two bodies relative to each other we demonstrate how it is possible to generate catheter deformation in excess of 90° despite this fundamental limitation of the actuating platform (the MRI scanner).”

Figure 1: Our flat magnetic ring in a fixed B_0 (light blue arrows) is rotated counter-clockwise from **A-G**. The most Northerly point on the ring will always be the North pole and, as such, the system wants to align this pole with the direction of B_0 (magnetic torque τ shown as green arrows). In **A** and **E**, $\theta = 0^\circ$ therefore $\sin(2\theta) = 0$ however, this represents an unstable equilibrium (an inverted pendulum). In **C** and **G**, $\theta = 90^\circ$ so $\sin(2\theta) = 0$. As we rotate from **D**, through **E** to **F** the North pole flips from one end of the body to the opposite. It is clear from this that θ can never exceed 90° .

3. It's claimed by the authors that the outer diameter of the proposed concentric tube catheter is 3 mm, 1.5 mm, and 0.4 mm. However, in the supplementary video, the size of the catheter seems relatively large, very different from the conventional magnetic catheter. Can the authors supplement the measurement figures of the catheter by using measurement tools? It's doubtful that such a large size can navigate through small vessels. Please explain this.

The reviewer rightly observes that the maximal diameter of the catheter (currently 4.0mm) is larger than standard catheters. This is, however, a proof of concept publication to demonstrate the design principle, modeling and operation of the CoSMA. Notwithstanding this, there is some reasonable argument that even at this scale, the increased dexterity and reduced material stiffness

of our design can offer some advantages over the current state of the art. Nevertheless, our next, and most pressing, activity is to miniaturize the design. Something which will both increase reachable space and reduce the size of the signal void. This has been limited, thus far, by the need to (a) transfer torsional stiffness through the catheter and (b) the need to be MR compatible. Stainless steel helically reinforced Pebax tubes can be commercially purchased at the millimeter to sub-millimeter diameter scale which conform to requirement (a) but not (b) and un-reinforced Pebax tubes at the same scale conform to requirement (b) but not (a). We are in discussions with both commercial manufacturers and fabrication/textiles research groups to modify the same process of reinforcement to employ MR compatible material (nylon is the most likely material). This is an ongoing work but the manufacturing process already exists so the research risk is low. We have modified the introduction to be clear that this isn't the target scale of manipulator but a proof of concept and updated our Conclusion and Discussion to reflect the ongoing manufacturing situation. We have also updated Fig. 14 and supporting video S4 to give a clearer view of the current scale of the catheter as well as adding scale bars to Videos S2 and S3.

Line 70 of the revised paper: “Even at the present scale, the increased dexterity and reduced material stiffness of our design can offer advantages over the current state of the art. Nevertheless, further miniaturization of the design will both increase reachable space and reduce signal void size.”

Line 278 of the revised paper: “Our next activity is to miniaturize the design, something which will both increase reachable space and reduce signal void size.”

4. The MR magnetic field is very strong, and it is very difficult to control the field to actuate a robot inside. The authors should elaborate on the control method of the MR magnetic field to ensure the controllability of the magnetic catheter in such a strong field. How is the control accuracy of the magnetic catheter using the MR magnetic field compared to the normal permanent/electromagnetic field?

The operator has no control over the magnetic field B_0 in the bore of the scanner. It is possible to manipulate magnetic bodies by leveraging the imaging gradients of the scanner (demonstrated by our group [5] and others e.g. [6], [7] however, gradient pull force is extremely small and the change in B_0 induced by the imaging gradients on a clinical MR system is negligible. The realization that this gradient force is so tiny is what motivated our use of the torque associated with B_0 . Based on the MR specific constraints outlined above we have modified the pre-existing concentric tube design such that the control inputs we now use are (as with mechanical concentric tubes) the base configurations of the catheter.

Controlling such a design comes with its own challenges which have a lot in common with the control challenges of elastic concentric tubes (in particular, [8] and [9]) but also rely on formulation of the magnetic energy/torque ([10]). In this article we make no attempt to perform formal control, the catheter is

manually actuated, all the modeling is performed offline, prior to navigation. We determined the design of the catheter and the necessary base poses for each step of each navigation (Section 5.5 of the submission) through simulation. The navigation itself is performed manually and not formally controlled in real-time. As this is a proof of concept of the design, the formal controllability of the catheter remains unaddressed but from our inquiries it appears this will be possible through the well encoded channels presented in the early concentric tube publications (i.e. Section IV. of [8]). Indeed, in terms of stability (a fundamental requirement for controllability) our design offers significant improvement over traditional elastic concentric tubes due to the vastly improved ratio of GJ/EI (see Section 5.2.2), a well documented limiting factor regarding the snap-through instability.

In terms of a controllability comparison to traditional magnetic actuation, for tip-driven/axially magnetized catheters this is effectively a solved problem (e.g. [11]) but for multi-DoF (active shape forming) catheters such as this, there are limited successful studies, one significant example being [12]. The principle of how we could control our system is almost identical to the theory presented in [12]. Deformation is a balance between magnetic and elastic energy, the enormous size of B_0 is no issue as the torque is a function of volume and geometry of the magnetic ring, which is itself tailored to maintain the elasto-magnetic equilibrium. The only serious complication to any such control model comes, as mentioned above, in the snap-through instability which is well understood and our design offers significant improvement on. Nevertheless, we acknowledge that we haven't attempted to formulate a control model and that this remains a significant future challenge. We have elaborated this point in the principle of operation section, and on the point specifically about controllability, in the Conclusion and Discussion as referred to in Query 6 (below).

Line 92 of the revised paper: “We cannot directly use the stationary background field (B_0) for actuation in any controllable direction, only the provision of actuating energy, we therefore must provide directional actuation through some alternative mechanism. Through rotation and translation of the respective sleeves we can create a variety of shapes which ultimately enable tortuous navigation (Fig. 1). Thus, the control inputs become the base configurations of the catheter as opposed to, as with traditional magnetic catheter designs, the actuating field settings.”

5. How can the magnetic field actuate the desired tube while keeping the other tube stationary to ensure the accuracy of the distal tube, as they are all affected by the magnetic field? This is a very critical issue for a magnetic catheter integrated with multiple magnetic components.

This is a good question and highly relevant for any manipulator featuring multiple, linearly dependent, active segments. When one tube is translated or rotated, the change in torque affects the pose of the other tube which then affects the magnetic torque of both. This is a conceptually similar problem to the

Figure 2: Screenshots from the simulation described in Section 5.5 (of the original submission). As Sleeve B is rotated, Sleeve C can be observed to be unaffected, as reflected in Supplementary Video S2.

model we developed in [13] for remnant shape-forming catheters. It is inherently impossible to hold tube C stationary whilst moving tube B. It is therefore necessary to move both tubes to a different equilibrium state whereby the pose of tube C remains the same. This revised balance has to be determined through simulation. In Section 5.5 we developed a model which accommodates these superimposed magnetic and elastic torques of the different tubes allowing us to accurately counteract this interplay. This model fundamentally relies on an iterative numerical solution to Equation 1, which is slow (0.5-10 seconds depending on desired shape). In order to control this in real-time our model will need to be improved to process a combination of current pose and a linearized approximation of this torque balance for a small forward time-step.

In practice, for this particular design, as can be observed in supplementary Video S2 and in Fig. 2, the inbuilt imbalance of magnetic torques (Ring C has higher volume and n_r than Ring B - Section 5.5) means sleeve C is almost perfectly stationary as sleeve B rotates and translates. We have amended the modeling section (5.5) of the paper to more clearly reflect this fundamental axiom of multi-magnet designs as well as supplementing the Conclusion and Discussion section to address the control model issue as detailed in Query 6 below.

Line 492 of the revised paper: “For any manipulator featuring multiple, linearly dependent, active segments, when one sleeve is translated or rotated, the change in torque affects the pose of the other sleeves which in turn affects the magnetic torque of both. This is a conceptually similar problem to the model we developed in [13] for remnant shape-forming catheters. Sleeve C will always move in response to movement of sleeve B. It is therefore theoretically necessary to move both tubes to a different equilibrium state whereby the pose of tube C remains the same. In practice, for this particular design, as can be observed in supplementary Video S2, the inbuilt imbalance of magnetic torques means sleeve C is almost perfectly stationary as sleeve B rotates and translates.”

6. One major motivation for an MR-compatible robot is the utilization of MR images as the closed-loop feedback to form a closed-loop controller. Have the authors considered this issue during the design process?

Yes, the ambition for this work is a catheter which navigates autonomously using the information provided by the MR scanner. Closed loop control using the MR scanner as the feedback sensor has been demonstrated in [14], [15]. These contributions prove the feasibility of the implementation of MR data as feedback at sufficient speed for control using untethered capsules. The principle difference with our contribution is that a soft catheter system requires a more sophisticated control model. We require a model which can take post-processed image information as pose error and establish 5 DoF incremental forward actuation information which will close this error. In terms of potential controllability, (a) the design of the catheter itself has a deliberate imbalance of magnetic torques ($\tau_C > \tau_B > \tau_A$, Section 5.5), (b) the catheter has a deliberately high GJ/EI ratio (Section 5.2.2) and (c) the model solves the torque equilibrium for all magnetic and elastic components to give full shape. These features provide a solid foundation for the significant future work of developing a real-time control model as discussed in point 5 above. We have expanded on this point in the Conclusion and Discussion to both acknowledge the current absence of closed loop control and to clarify that the system design is suitable for this development.

Line 259 of the revised paper: “Controlling such a design comes with its own challenges which have a lot in common with the control challenges of elastic concentric tubes ([40], [41]) but also rely on formulation of the magnetic energy and torque ([32]). In this article we make no attempt to perform formal control, the catheter is manually actuated, and all the modelling is performed offline, prior to navigation. As this is a proof of concept of the design, the formal controllability of the catheter remains unaddressed but from our inquiries it appears this will be possible through the well encoded channels presented in the early concentric tube publications. Indeed, in terms of stability (a fundamental requirement for controllability) our design offers significant improvement over traditional elastic concentric tubes due to the vastly improved ratio of GJ/EI (Section 5.2.2), a well documented limiting factor regarding the snap-through instability. In terms of a controllability comparison to traditional magnetic actuation, for tip-driven/axially magnetized catheters this is effectively a solved problem (e.g. [42]). For multi-DoF (active shape forming) catheters such as this, the principle of how we could control our system is almost identical to the theory presented in [43].”

7. It’s recommended that the authors should revise the abstract, removing (a) and (b), and replace them with “first” and “second” instead. We have made this change, highlighted in red in the revised version.

8. It’s recommended that the author should give the full name of abbreviations when first using them, e.g., MRI-“Magnetic Resonance Imaging”. The authors should use the term “MR” for many locations throughout the paper instead of “MRI”, e.g., in Figure 1, the MRI field should be considered revised

to “MR Field”.

We have made these changes, highlighted in red in the revised version.

9. The figures should be further improved and edited, considering the standard of the current publications, e.g., <https://doi.org/10.1038/s44172-025-00424-3>

We understand the reviewers concern about the quality of the figures. Notwithstanding that a large number of the images were taken on MR compatible endoscopic cameras and are thus, by necessity, low resolution. We have upgraded Figs 5, 6, 8, 9, 10, 12, 14 and Tables 1-5.

10. In Supplementary Video S4, the fabrication process illustration is shown. However, it is recommended that the authors supplement the real picture of the proposed catheter in the video at every fabrication stage.

We have updated the fabrication figure (Fig. 14) and the fabrication video (Supplementary Video S4) with microscope still-frames of the three components of the CoSMA.

2 Reviewer 2 Comments:

This study present a structurally adaptable CoSMA, constructed of two sleeves and one guidewire, driven by the background field of the MRI scanner. It enables multi-degrees of freedom motion. The mathematical simulation and phantom test were performed. However, there are several points need to be addressed in the calculation and discussion.

Thank you to the reviewer for bringing up some very interesting queries on the clinical feasibility of the design. We hope that these responses address your concerns satisfactorily.

1. This design enables multi-degree of freedom, but it is difficult to operate for radiologist. The magnetic field is perpendicular to the central axis of the sleeves. The radiologist needs to consider when should the second sleeve be placed and even the third tip.

The current design is manually actuated and was driven under visual feedback, i.e. the operator manipulates five degrees of freedom - the insertion position of the three sleeves and the rotation of sleeves B and C - sleeve A is axially symmetrical so rotation is redundant. This is performed using two hands whilst watching a camera feed. This sounds tricky and it is difficult to convey in a paper the experience of operating the catheter but the individual sleeves don't move unless they are manipulated from the base, i.e. when all the bases are completely released, the catheter will remain wherever it is. Each individual DoF can be actuated with one hand whilst the others are held still with the other hand without any skill/dexterity issues. The magnetic field is homogeneous and

immutable which means the operator quite quickly becomes accustomed to how the different rings (and the tip) will deform. As discussed in Section 3.2, the navigational repeats were timed at a mean of $138s \pm 45s$ across all paths for completely unskilled/untrained operators. Indeed, it was a pleasant surprise to the authors how straightforward navigation became once the design and construction had been iteratively refined. The challenge in moving beyond this approach is the rapid harvesting and post-processing of the MR images in such a way that the visual feedback can be replaced with a digital reconstruction of the catheter’s shape and pose which is equally as useful to the operator as the multiple camera angles. This is a subject we now address more thoroughly in the Conclusion and Discussion section.

Line 243 of the revised paper: “The rapid harvesting and post-processing of MRI information such that this potential wealth of sensory information can be usefully implemented by an operating model. Optimization of imaging methods to focus on scan time reduction, dedicated undersampling and/or reconstruction techniques such as parallel imaging, compressed sensing and/or AI-based acceleration methods, which are available on clinical MR systems, can be employed.”

More ambitiously, we are also working to automate the process such that the base configurations will be motorized and the navigation can run either (a) from an operator on a control pad where tip and body pose are more intuitively controlled or (b) under closed loop control without the real-time input of an operator (with the operator present as a fail-safe measure). Whilst the reviewer raises a valid concern, it is important to note that, at this early level of technological readiness, optimization of the user experience is a secondary concern behind demonstration of the functionality of the design. We have elaborated on this point in the Conclusion and Discussion section.

Line 254 of the revised paper: “The operational hardware of the CoSMA (control of the position and orientation of the various sleeves) must be automated to allow actuation in response to feedback. This relies on a 5 DoF, MR compatible motor arrangement. Currently, the system is manually operated based on pre-planned base configurations. The current numerical model will need to be faster and more robust before it can offer a path to a closed-loop control solution ...”

2. If the operation was wrong, and I think it can be easily wrong, can it damage the surround vessel? The worst situation needs to be considered.

In Fig. 7 (of the original submission) we discuss the favorable material softness of our design compared to elastic concentric tube catheters and the manually operated clinical standard. We have added a comparison of these values with various human tissues below and in Section 5.2 to add clarity to this comparison. Clearly, the new robot design is still stiffer than the more sensitive

anatomical regions but this work represents a significant improvement on the state of the art. The maximum torque our system can generate is $\approx 1mNm$

Material/Design	Elastic Modulus (Nm^{-2})
This work, Sleeves B and C	$500kPa$
This work, Sleeve A	1×10^3kPa
Elastic Concentric Tube Robots	50×10^6kPa
Manual Sheaths/Guidewires	$8 \times 10^3 - 500 \times 10^3kPa$
Arterial Tissue [16]	$160 - 360kPa$
Lung Tissue [17]	$2.5 - 4kPa$
Brain Tissue [18]	$0.5 - 1kPa$

Table 1: Comparison of Elastic (Young’s) modulus of various catheter designs and anatomical tissues.

(Table 1 - the 4.0 mm ring at $\theta = 45^\circ$ to B_0). This is equivalent, at a catheter radius of 1.5mm, to a force of 0.7N. This is comfortably below the upper limit of 2.5N reported in [19] (for urinary catheters) and comparable to the 0.5N reported in [20], the cable operated cardiovascular robotic catheter, specifically designed to minimize contact force during navigation and the 0.6N in the comparable system in [21].

There is also the issue of the rigid and potentially sharp edges of the metal hoop within delicate anatomy. This can be easily mitigated using laser cut and burred rings and a thin layer of high stiffness elastomer (e.g. PDMS) as a coating. These last issues are downstream manufacturing considerations for clinical readiness but nonetheless, we have updated our discussion to include this important point.

Line 333 of the revised paper: “Table 1: Comparison of Elastic (Young’s) modulus of various catheter designs against various anatomical tissues. The present contribution is still stiffer than the more sensitive anatomical regions but represents a significant improvement on the state of the art.”

Line 276 of the revised paper: “delicate anatomy must be protected from the rigid and potentially sharp edges of the metal hoop. Laser cut and burred rings with a thin layer of high stiffness elastomer (e.g. PDMS) can mitigate both of these issues.”

3. The distortion or image voids need to be presented when all three components were placed out. I am afraid that the image voids are bigger than the diameter of big vessels around the heart. The size of such big tip limits the application in small vessels.

This is a prescient point that has been dominant in MR actuated robotics since the inception of the subject. Signal voids are significantly larger than the magnetic body which is causing them and thus cause an unavoidable blackout

in the anatomical region of interest. By harnessing the torque associated with B_0 , as opposed to using imaging gradients to impart force on a ferrous sphere as previous works have done (e.g. [6]), we have significantly reduced the required volume of metal and thus the surrounding signal void.

Our high-field scanner ($B = 7T$) is capable of producing field gradients $\nabla B = 660mTm^{-1}$. In this scanner, our largest ring (4.0mm diameter gives 0.6 mm^3 volume) produces a peak torque of $\tau_{mag} \approx 1mNm$. From linear Bernoulli beam theory, for a 20mm catheter, this produces equivalent deformation to a tip point-force of $0.05N$. According to $F = \nabla B \cdot m$ [5], to produce this force would require a 2.4mm diameter sphere of the same material as our rings. Approximately 100 times the volume of our ring with the attendant increase in void size.

Beyond this, our future work consists of (a) reducing the overall size of the catheter which will inevitably reduce the signal void size, (b) further optimization of the imaging sequence to reduce this void (dedicated undersampling and / or reconstruction techniques, such as parallel imaging, compressed sensing and / or AI-based acceleration methods, which are available on clinical MR systems.), and (c) reconstruction of the anatomical background from MR data harvested prior to the presence of the signal void to compensate for the "missing region". Clearly this last point will not be able to capture the effects of any anatomical contact occurring within the signal void, but if we can make the void sufficiently small, we should be able to infer any anatomical deformations (We discuss image reconstruction further in response to the next query). We have added further detail to the discussion to reflect this concern.

Line 243 of the revised paper: "The rapid harvesting and post-processing of MRI information such that this potential wealth of sensory information can be usefully implemented by an operating model. Optimization of imaging methods to focus on scan time reduction, dedicated undersampling and/or reconstruction techniques such as parallel imaging, compressed sensing and/or AI-based acceleration methods, which are available on clinical MR systems, can be employed."

Line 274 of the revised paper: "fabrication of the CoSMA should be both automated and miniaturized. The manufacturing and positioning of the magnetic elements currently represents a significant source of operational error. Furthermore, delicate anatomy must be protected from the rigid and potentially sharp edges of the metal hoop. Laser cut and burred rings with a thin layer of high stiffness elastomer (e.g. PDMS) can mitigate both of these issues. Our next activity is to miniaturize the design, something which will both increase reachable space and reduce signal void size."

4. It is difficult to localize or track such tips with three component using MRI technique.

This is a very good point. At present the images shown are of three signal voids spaced out. For live tracking there will be poses where the voids are superimposed in the same physical space. In some cases a void may be completely

obscured by a larger void in the same 3D location. We will, however, know the base configurations of the three sleeves and the poses of the three sleeves at an incrementally earlier time step when we had a cleaner image. From (a) a noisy image, (b) base configuration information coupled with a physics model, and (c) previous pose information (“image flow”) it should be possible to reconstruct the current pose of the three sleeves. This is something which remains untested at present as we consider this paper to be essentially a proof of concept of the design principle. Of course, as with Query 3 (above), by reducing the absolute size of the image void through both miniaturization of the system and scanner setting optimization (See Query 3 (above) and Reviewer 3 Query 2), this issue will be reduced, however it still remains to be addressed. We have significantly modified the Discussion section to consider this point as it represents a potential future barrier to clinical relevance. As a further mitigation, if required, there is precedent for the use of fiducial markers in tracking of MR robotics [22]. Whilst this would add complexity to the fabrication process and size to the complete catheter, it remains an alternative solution should pose reconstruction from signal voids prove infeasible.

Line 247 of the revised paper: “At present the images shown are of three distinct signal voids. For live tracking there will be poses where the voids are superimposed in the same physical space. In some cases a void may be completely obscured by a larger void in the same 3D location. We will, however, know the base configurations of the three sleeves and the poses of the three sleeves at an incrementally earlier time step when we had a cleaner image. From a noisy image, base configuration information coupled with a physics model, and previous pose information (“image flow”) it should be possible to reconstruct the current pose of the three sleeves.”

3 Reviewer 3 Comments:

This work introduces a concept of MRI guided concentric tube catheter based on concentrically actuated sleeves with iron rings wrapped around their outer surface to produce magnetic torques to bend the sleeves based on the interaction between the iron rings and the principle field of the MRI. Although the demonstration provided here is preliminary and still far from clinical translation, the concept is new and worth exploring. This submission provides a solid grounding for further exploring this concept in the future. Despite this promising proposition, some aspects of the work remain unclear, and I have some concerns regarding the feasibility of this concept in more realistic settings which I detail below.

We thank the reviewer for their considered and useful input. We address their concerns below.

1. The experimental demonstration was performed in a preclinical 7T MRI

machine. It is not clear whether the simulated scenario are also restricted to this principal field value. As this property influences both the artifacts, acquisition time, and dexterity of the device, I think it is important to discuss (at least in simulation/theory) how the proposed design translates to clinical systems (0.5 – 3T).

The simulation was indeed performed for a magnetic field strength $B_0 = 7T$. The magnitude of the magnetic torque, subject to the background field exceeding the saturation magnetization, is invariant with increase in field:

$$\tau_{mag} = \frac{1}{2}\mu_0 v |n_r - n_a| m_s^2 \sin(2\phi), \quad (2)$$

Saturation magnetization $m_s = 1.43 \times 10^6 (A/m)$ for annealed iron. $H_0 = \frac{B_0}{\mu_0}$ thus, saturation occurs at 1.8T background field. Below 1.8T, for annealed iron components, the assumption of saturation is violated and the torque calculation becomes [10]:

$$\tau_{mag} = \frac{1}{2n_r n_a} \mu_0 v |n_r - n_a| H^2 \sin(2\theta), \quad (3)$$

which scales quadratically with background field (for the purposes of this discussion $\phi \approx \theta$).

The signal void is caused by local field perturbation, itself a function of the magnetization of the metallic component. This magnetization is induced entirely by B_0 and follows the standard hysteresis curve. This can be approximated as a linear increase according to the material's susceptibility ($\chi = 5000[-]$ www.hyperphysics.phy-astr.gsu.edu), then saturating at the same B_0 [10].

The effect of background field on artifact area was analyzed in detail in [23] for titanium and stainless steel implants using a variety of gradient-recalled echo and spin echo sequences. A mean equivalence is presented as:

$$\frac{A_{B_{01}}}{A_{B_{02}}} = \sqrt{\frac{B_{01}}{B_{02}}}, \quad (4)$$

for B_0 in the range 1.5T to 7T where $A_{B_{0i}}$ is the area of the signal void associated with B_{0i} .

It is evident from Fig. 3 that decreasing B_0 from 7T will improve the image (reduce artefacts) without loss of torque and therefore manipulability. This is true down to 1.8T where the relationship becomes more complex and unpredictable but it is reasonable, based on Fig.3, to assert that overall performance in a 1.5T clinical scanner would be improved over the 7T scanner. This may well not be true for lower field scanners. It is worth noting that this 1.8T threshold is lower for materials with lower saturation magnetization (e.g. Nickel saturates below 1T). By using such materials the curves in Figure 3 move to the left with both reduced torque and reduced artefact areas. This loss of torque can be compensated with an increase in metallic volume, inducing a linearly commensurate increase in artefact area thus permitting comparable performance in lower field

Figure 3: Background field (B_0) against magnetic torque [10] (red), artefact area [23] (blue) and magnetization [10] (green). All variables normalized against results in the 7T scanner used in this work.

scanners, at least theoretically. We have no evidence that we are Signal-to-Noise-Ratio (SNR) limited on the present system and no reason to believe this would be the case down to 1.5T background fields so acquisition time should be unaffected by any transfer to a clinical MR scanner. We have revised Section 5.4 and included the above figure in Figure 10 of the methods to reflect this response.

Line 446 of the revised paper: “The magnitude of the magnetic torque, subject to the background field exceeding the saturation magnetization, is invariant with increase in field. The effect of background field on artifact area was analyzed in detail in [23]. A mean equivalence is presented as:

$$\frac{A_{B_{01}}}{A_{B_{02}}} = \sqrt{\frac{B_{01}}{B_{02}}}, \quad (5)$$

for B_0 in the range 1.5T to 7T where $A_{B_{0i}}$ is the area of the signal void associated with B_{0i} . It is evident from Fig. 10 E that decreasing B_0 from 7T will improve the image (reduce artefacts) without loss of torque and therefore manipulability. This is true down to 1.8T where the relationship becomes more complex and unpredictable but it is reasonable to assert that overall performance in a 1.5T clinical scanner would be improved over the 7T scanner.”

2. I appreciate the efforts of the authors to comment on the artifacts and discuss the value of the Feret diameter. That being said, it is clear that the use of the iron ring present a fundamental limitation in terms of acquisition time and image quality and artifacts. While these limitations are acknowledged, the authors do not really discuss how these limitations could be overcome in practice to make this concept clinically realistic (i.e. to ensure acquisition time in the sec-

ond rather than the minute range, and whether the artifacts are expected to be reduced significantly enough to make them acceptable for the target procedure).

We thank the reviewer for raising this point. The aim of our current work was to visualize the robot as accurately as possible rather than to optimize the imaging methods or to focus on scan time reduction. Indeed, no dedicated undersampling and / or reconstruction techniques, such as parallel imaging, compressed sensing and / or AI-based acceleration methods, which are available on clinical MR systems, have been employed. Furthermore, the artefacts caused by the iron ring will be reduced at lower field magnetic strengths. It is also worth noting that, by harnessing the torque associated with B_0 , as opposed to using imaging gradients to impart force on a ferrous sphere as previous works have done (e.g. [6]), we have significantly reduced the required volume of metal and thus the surrounding signal void. Our high-field scanner ($B = 7T$) is capable of producing field gradients $\nabla B = 660mTm^{-1}$. In this scanner, our largest ring (4.0mm diameter gives 0.6 mm^3 volume) produces a peak torque of $\tau_{mag} \approx 1mNm$. From linear Bernoulli beam theory, for a 20mm catheter, this produces equivalent deformation to a tip point-force of $0.05N$. According to $F = \nabla B \cdot m$ [5], to produce this force would require a 2.4mm diameter sphere of the same material as our rings. Approximately 100 times the volume of our ring with the attendant increase in void size. We consider optimization of the imaging approach to be beyond the scope of this work and subject to future research. To address the reviewer’s comment, we have added this point to the Conclusion and Discussion section.

Line 243 of the revised paper: “The rapid harvesting and post-processing of MRI information such that this potential wealth of sensory information can be usefully implemented by an operating model. Optimization of imaging methods to focus on scan time reduction, dedicated undersampling and/or reconstruction techniques such as parallel imaging, compressed sensing and/or AI-based acceleration methods, which are available on clinical MR systems, can be employed.”

3. Thermal effects are not discussed: can the authors elaborate on what is the expected heat and possible temperature elevation in the iron rings produced by the RF pulses?

This is a good question, indeed, there are works exploiting this very effect either via current carrying loops [24] or via nano-particulate swarms [25, 26] but these pertain to completely different conceptual designs and geometries. The effects of heating on metallic implants and devices due to gradient and resonant fields are well explored [27] with, at times, significant heating reported, particularly resonant heating in long, slender wires [28]. Heating of significance ($> 1^\circ C$ locally [29]) occurs when the wire’s length approaches or exceeds $1/4$ of the RF wavelength (<https://mriquestions.com/gradient-vs-rf-heating.html>) which, for a 1.5T scanner, would be $\approx \frac{1}{4} \times 500mm = 125mm$, and for a 7T

scanner $\approx 26.8mm$, in both cases considerably longer the characteristic length of our rings. In terms of analysis of the unwanted heating effect in metallic implants (lower aspect ratio, higher volume), [30] showed an increase of $< 2.5^\circ C$ after 6 minutes scan time in a 1.5T and a 3T scanner for Cobalt-Chromium-Molybdenum alloy hip implant (note - this result was also shown to be a function of the scan sequence i.e spin-echo, gradient echo, echo-planar). Given the low bulk volume and relatively low aspect ratio of our metallic rings, it is highly unlikely that we fall into either of the above categories. We further argue that our catheter is designed for endovascular interventions, where the flowing blood will dissipate heat. It is therefore reasonable to conclude that our design is not susceptible to dangerous heating during scan-time.

There is an entire (small) field of research harnessing the magnetic field of the scanner for robotic actuation, similar to this work, summarized in the review paper [29]. Due to the low state of technological readiness of these robots, nobody has yet satisfactorily addressed this issue of potential unwanted heating. “As the wavelength of the RF waves can be of the order of the sizes of those metallic materials/devices, not only magnetic but also electrical coupling needs to be considered, such as in the case of metallic rods turning into antennas. In such cases, extreme heating spots can form and harm the patient” [29]. As mitigation, in the event that we do fall into either (or both) of the above categories subject to dangerous unwanted heating effects there is a well developed research field in soft catheters exploiting materials with temperature dependent stiffness (e.g. [31]). These catheters exhibit thermal surface insulation to isolate the anatomy from large bulk material temperature changes ($80^\circ C$ catheter core temperature, $49^\circ C$ catheter surface temperature, insulation thickness $\approx 0.5mm$). Such insulation could perform the same role in our system.

The system characteristics of our pre-clinical scanner are completely different to a clinical scanner, e.g. while the Larmor frequency is significantly higher ($\approx 300MHz$ in the 7T scanner, $\approx 64MHz$ for a 1.5T clinical scanner) and the wavelength correspondingly shorter, the power requirements for RF pulses are considerably lower (typically 10’s to 100’s of W in our pre-clinical 7T scanner compared to $\approx 1’s$ to $10’s$ of KW for a standard clinical scanner, the imaging sequence itself remains unconfirmed. As such we haven’t performed thermal tests on our specific sample. We have added this consideration to Section 5.4.

Line 453 of the revised paper: “Due to the low volume and characteristic length of our ferrous components, heating effects are most likely negligible but this is something that will require further investigation when we move to a clinical scanner.”

4. How are the sleeves insertion/retraction and roll actuated? The device exhibits 5 DOF, so I can see that being a challenge to design an actuation unit that is both compact enough and that does not deteriorate the image quality. It is said the “navigations were performed three times using the prototype catheter under visual feedback via manual operation of the base configuration”;

does that mean there was no motorized unit throughout the experiments?

The current prototype is a manually actuated proof of concept of the operating principle. It is surprisingly straightforward from both a dextrous and an intuitive perspective for a single individual to drive all 5 DoFs (although this statement is clearly subjective and unverifiable). There is also an appeal from a clinical perspective to a manually actuated system in terms of maintaining control in the hands of the clinician, as noted by reviewer 2 (query 1), this entire design could end up as a manually operated, softer and more dexterous, alternative to the current pre-bent sheath and guide-wire system.

As roboticists, we obviously want a 5 DoF motorized actuation unit as the next step. Initially this would be designed as a Graphical User Interface where the operator can drive tip and body pose from a control pad. The ultimate objective, as ever, is a closed loop controller where a desired path is entered and the navigation is performed, it is self-evident that this is significant work away. The major challenges to the future of this work, as the reviewer has noticed, are image acquisition and post-processing, design and fabrication of the catheter and a fit-for-purpose control model. In terms of the practical feasibility of a 5 DoF motorized unit, we currently have two MR compatible syringe pumps (PHD 22/2000, Harvard Apparatus, Syringe Pump Series) which we can operate from micro-controllers from the PC which runs the paravision MR imaging software. They are, however, bulky (163 x 228 x 279 mm) and five such drivers would not be practical. Alternative MR compatible compact drivers are available either commercially (e.g. [32] offer a 134x72x287mm digital unit) or academically (e.g. [33] demonstrated a pneumatically driven unit below 100mm in all three dimensions). We have updated the Conclusion and Discussion to reflect this practical concern.

Line 254 of the revised paper: “The operational hardware of the CoSMA (control of the position and orientation of the various sleeves) must be automated to allow actuation in response to feedback. This relies on a 5 DoF, MR compatible motor arrangement. Currently, the system is manually operated based on pre-planned base configurations.”

References

- [1] M. Richter, V. K. Venkiteswaran, and S. Misra, “Concentric tube-inspired magnetic reconfiguration of variable stiffness catheters for needle guidance,” *IEEE Robotics and automation letters*, vol. 8, no. 10, pp. 6555–6562, 2023.
- [2] Z. Li, J. Li, Z. Wu, Y. Chen, M. Yeerbulati, and Q. Xu, “Design and hierarchical control of a homocentric variable-stiffness magnetic catheter for multiarm robotic ultrasound-assisted coronary intervention,” *IEEE Transactions on Robotics*, vol. 40, pp. 2306–2326, 2024.

- [3] M. M. Schmauch, S. R. Mishra, E. E. Evans, O. D. Velev, and J. B. Tracy, “Chained iron microparticles for directionally controlled actuation of soft robots,” *ACS applied materials & interfaces*, vol. 9, no. 13, pp. 11 895–11 901, 2017.
- [4] H. Liu, X. Teng, Z. Qiao, H. Yu, S. Cai, and W. Yang, “A concentric tube magnetic continuum robot with multiple stiffness levels and high flexibility for potential endovascular intervention,” *Journal of Magnetism and Magnetic Materials*, vol. 597, p. 172023, 2024.
- [5] N. Murasovs, P. Lloyd, A. Bacchetti, Y. L. May, J. Armitage, O. Cespedes, E. Dall’Armellina, J. H. Chandler, J. E. Schneider, and P. Valdastri, “Gradient pulling of a tethered robot via a magnetic resonance imaging system,” *Device*, 2025.
- [6] M. E. Tiriyaki and M. Sitti, “Magnetic resonance imaging-based tracking and navigation of submillimeter-scale wireless magnetic robots,” *Advanced Intelligent Systems*, vol. 4, no. 4, p. 2100178, 2022.
- [7] O. Felfoul, A. T. Becker, G. Fagogenis, and P. E. Dupont, “Simultaneous steering and imaging of magnetic particles using mri toward delivery of therapeutics,” *Scientific reports*, vol. 6, no. 1, p. 33567, 2016.
- [8] P. E. Dupont, J. Lock, B. Itkowitz, and E. Butler, “Design and control of concentric-tube robots,” *IEEE Transactions on Robotics*, vol. 26, no. 2, pp. 209–225, 2009.
- [9] R. J. Webster, A. M. Okamura, and N. J. Cowan, “Toward active cannulas: Miniature snake-like surgical robots,” in *2006 IEEE/RSJ international conference on intelligent robots and systems*. IEEE, 2006, pp. 2857–2863.
- [10] J. J. Abbott, O. Ergeneman, M. P. Kummer, A. M. Hirt, and B. J. Nelson, “Modeling magnetic torque and force for controlled manipulation of soft-magnetic bodies,” *IEEE Transactions on Robotics*, vol. 23, no. 6, pp. 1247–1252, 2007.
- [11] J. Edelmann, A. J. Petruska, and B. J. Nelson, “Magnetic control of continuum devices,” *The International Journal of Robotics Research*, vol. 36, no. 1, pp. 68–85, 2017.
- [12] D. Lin, W. Chen, K. He, N. Jiao, Z. Wang, and L. Liu, “Position and orientation control of multisection magnetic soft microcatheters,” *IEEE/ASME transactions on mechatronics*, vol. 28, no. 2, pp. 907–918, 2022.
- [13] P. Lloyd, G. Pittiglio, J. H. Chandler, and P. Valdastri, “Optimal design of soft continuum magnetic robots under follow-the-leader shape forming actuation,” in *2020 International Symposium on Medical Robotics (ISMR)*. IEEE, 2020, pp. 111–117.

- [14] C. Bergeles, P. Vartholomeos, L. Qin, and P. E. Dupont, “Closed-loop commutation control of an mri-powered robot actuator,” in *2013 IEEE international conference on robotics and automation*. IEEE, 2013, pp. 698–703.
- [15] O. Erin, C. Alici, and M. Sitti, “Design, actuation, and control of an mri-powered untethered robot for wireless capsule endoscopy,” *IEEE Robotics and Automation Letters*, vol. 6, no. 3, pp. 6000–6007, 2021.
- [16] P. Nabeel, J. Joseph, M. I. Shah, M. Sivaprakasam *et al.*, “Measurement of arterial young’s elastic modulus using artsens pen,” in *2018 IEEE International Symposium on Medical Measurements and Applications (MeMeA)*. IEEE, 2018, pp. 1–6.
- [17] S. R. Polio, A. N. Kundu, C. E. Dougan, N. P. Birch, D. E. Aurian-Blajeni, J. D. Schiffman, A. J. Crosby, and S. R. Peyton, “Cross-platform mechanical characterization of lung tissue,” *PloS one*, vol. 13, no. 10, p. e0204765, 2018.
- [18] N. D. Leipzig and M. S. Shoichet, “The effect of substrate stiffness on adult neural stem cell behavior,” *Biomaterials*, vol. 30, no. 36, pp. 6867–6878, 2009.
- [19] X. Ling, M. Tradewell, R. M. Sweet, and T. M. Kowalewski, “A catheter insertion force assessment tool: design and preclinical results,” in *Engineering and Urology Society 33rd Annual Meeting. San Fransisco, CA*, 2018, p. 28.
- [20] C. Li, “Catheter modelling and force estimation in endovascular application,” Master’s thesis, University of Twente, 2023.
- [21] Y. Wang, S. Guo, N. Xiao, Y. Li, and Y. Jiang, “Online measuring and evaluation of guidewire inserting resistance for robotic interventional surgery systems,” *Microsystem Technologies*, vol. 24, no. 8, pp. 3467–3477, 2018.
- [22] E. E. Tuna, N. L. Poirot, J. B. Bayona, D. Franson, S. Huang, J. Narvaez, N. Seiberlich, M. Griswold, and M. C. Çavuşoğlu, “Differential image based robot to mri scanner registration with active fiducial markers for an mri-guided robotic catheter system,” in *2020 IEEE/RSJ International Conference on Intelligent Robots and Systems (IROS)*. IEEE, 2020, pp. 2958–2964.
- [23] T. Spronk, O. Kraff, J. Kreutner *et al.*, “Development and evaluation of a numerical simulation approach to predict metal artifacts from passive implants in mri. magn reson mater phy,” 2021.
- [24] T. Niwa, Y. Takemura, T. Inoue, N. Aida, H. Kurihara, and T. Hisa, “Implant hyperthermia resonant circuit produces heat in response to mri unit radiofrequency pulses,” *The British journal of radiology*, vol. 81, no. 961, pp. 69–72, 2008.

- [25] A. Shakeri-Zadeh and J. W. Bulte, “Imaging-guided precision hyperthermia with magnetic nanoparticles,” *Nature reviews bioengineering*, vol. 3, no. 3, pp. 245–260, 2025.
- [26] M. Wabler, W. Zhu, M. Hedayati, A. Attaluri, H. Zhou, J. Mihalic, A. Geyh, T. L. DeWeese, R. Ivkov, and D. Artemov, “Magnetic resonance imaging contrast of iron oxide nanoparticles developed for hyperthermia is dominated by iron content,” *International Journal of Hyperthermia*, vol. 30, no. 3, pp. 192–200, 2014.
- [27] L. Winter, F. Seifert, L. Zilberti, M. Murbach, and B. Ittermann, “Mri-related heating of implants and devices: a review,” *Journal of Magnetic Resonance Imaging*, vol. 53, no. 6, pp. 1646–1665, 2021.
- [28] M. K. Konings, L. W. Bartels, H. F. Smits, and C. J. Bakker, “Heating around intravascular guidewires by resonating rf waves,” *Journal of Magnetic Resonance Imaging*, vol. 12, no. 1, pp. 79–85, 2000.
- [29] O. Erin, M. Boyvat, M. E. Tiryaki, M. Phelan, and M. Sitti, “Magnetic resonance imaging system-driven medical robotics,” *Advanced Intelligent Systems*, vol. 2, no. 2, p. 1900110, 2020.
- [30] A. Arduino, U. Zanollo, J. Hand, L. Zilberti, R. Brühl, M. Chiampi, and O. Bottauscio, “Heating of hip joint implants in mri: the combined effect of rf and switched-gradient fields,” *Magnetic Resonance in Medicine*, vol. 85, no. 6, pp. 3447–3462, 2021.
- [31] Y. Piskarev, Y. Sun, M. Righi, Q. Boehler, C. Chautems, C. Fischer, B. J. Nelson, J. Shintake, and D. Floreano, “Fast-response variable-stiffness magnetic catheters for minimally invasive surgery,” *Advanced Science*, vol. 11, no. 12, p. 2305537, 2024.
- [32] Simutec, “Mri compatible multimodality motion stage v2,” 2025.
- [33] F. S. Farimani and S. Misra, “Introducing pneuact: Parametrically-designed mri-compatible pneumatic stepper actuator,” in *2018 IEEE international conference on robotics and automation (ICRA)*. IEEE, 2018, pp. 200–205.

An MRI Steered and Imaged Concentric Tube Catheter for Endoluminal Interventions - Authors Response

Manuscript ID: COMMSENG-25-0803-T

We again thank the reviewers and editors for their input to enhance our proposed publication. As with the previous round, changes to the submission appear as **red text**. In the present response we report comments from reviewers in **green text** and the response from the authors with **blue text**.

Reviewer #1 (Remarks to the Author): This paper presents a structurally adaptable “Coaxial Sleeve Magnetic Actuator” (CoSMA), driven by the background field of the MRI scanner. The authors have addressed most of my comments in the revised manuscript. However, there are still several concerns before considering publications. Please see the detailed comments below:

Thank you for taking the time to review our article again. I hope the below comments and changes address your concerns satisfactorily.

1. The authors should consider revising the title of the paper, as the current title does not reflect the characteristics of the proposed robot. The targeted surgery could also be included in the title to enhance the functionality of the proposed robot.

Thank you for your feedback. We have modified the title to: “**An MRI Steered and Imaged Concentric Tube Catheter for Endoluminal Interventions**”. We feel this title is a clearer reflection of the operating principles and potential use cases of the contribution.

2. It is recommended that the authors provide a control block diagram including the input/output, MR field control, and MR imaging feedback in the Method section.

We have addressed this recommendation by adding a new figure (Fig. S8) that presents a control block diagram of the system. The figure and accompanying caption describe the operating methodology of the current prototype. Automation using rapid MR sensory feedback and a closed-loop control model represents a natural next step for this system; however, the present work focuses on demonstrating the operating principle and experimental feasibility of the catheter.

Figure S8: (A-D) Four sample insertion steps of the CoSMA as controlled by the process in (E).

Desired pose derives from the preoperative scan centre-lines updated for the current state of insertion (catheter length). Initially, all sleeves are contracted together (tips coincident), translation of one or more sleeves moves to a new desired pose in accordance with Table S5. Error is closed via rotation of respective sleeves before the subsequent translation step is initiated. In the current work, the phantom is transparent and feedback comes from MR compatible cameras. In future works, after optimization of the MR sequence, feedback will be supplied from the MR Images and a controller can be implemented from this data.

3. The authors should give a clear description of the design of the proposed magnetic catheter using a figure including the advancement mechanism. The outer diameter of each tube should be clearly labelled.

We have added a new figure (Fig. S8) that includes a block diagram and caption describing how advancement is determined (based on visual feedback). In the current prototype, advancement is implemented manually, as outlined in the Conclusion and Discussion. This reflects the proof-of-concept nature of the system and allows direct operator control during evaluation.

In order to be clear about the physical scale of our design, we have appended dimensions to Figure S9.

4. The figures should be further improved and reedited to be consistent with the journal style.

There are now six figures in the main body of the article, the remaining figures appear in the supporting material document. We have endeavoured to improve Figures 5 and 6 and, furthermore, will be glad to work with the editorial staff during the production phase of the article (if accepted) and comply with their requests to bring our figures to the standards of the journal.

Reviewer #2 (Remarks to the Author):

The authors have fully addressed my comments. I have no further suggestions.

Thank you.

Reviewer #3 (Remarks to the Author):

The authors addressed all my comments and I have no further remarks or suggestions.

Thank you.